# Prefix-Tree Decoding for Predicting Mass Spectra from Molecules

**Samuel Goldman**
Computational and Systems Biology
MIT
Cambridge, MA 02139
samlg@mit.edu

**John Bradshaw**
Chemical Engineering
MIT
Cambridge, MA 02139
jbrad@mit.edu

**Jiayi Xin**
Statistics and Actuarial Science
The University of Hong Kong
Pokfulam, Hong Kong
xinjiayi@connect.hku.hk

**Connor W. Coley**
Chemical Engineering
Electrical Engineering and Computer Science
MIT
Cambridge, MA 02139
ccoley@mit.edu

## Abstract

Computational predictions of mass spectra from molecules have enabled the discovery of clinically relevant metabolites. However, such predictive tools are still limited as they occupy one of two extremes, either operating (a) by fragmenting molecules combinatorially with overly rigid constraints on potential rearrangements and poor time complexity or (b) by decoding lossy and nonphysical discretized spectra vectors. In this work, we use a new intermediate strategy for predicting mass spectra from molecules by treating mass spectra as sets of molecular formulae, which are themselves multisets of atoms. After first encoding an input molecular graph, we decode a set of molecular subformulae, each of which specify a predicted peak in the mass spectrum, the intensities of which are predicted by a second model. Our key insight is to overcome the combinatorial possibilities for molecular subformulae by decoding the formula set using a prefix tree structure, atom-type by atom-type, representing a general method for ordered multiset decoding. We show promising empirical results on mass spectra prediction tasks.

## 1 Introduction

As the primary tool to discover unknown small molecule structures from biological samples, tandem mass spectrometry (MS/MS) experiments have enabled the identification of numerous important molecules implicated in health and disease [4, 38, 51]. Tandem mass spectrometers are capable of isolating, fragmenting, and measuring the resulting fragment masses of small molecules from a sample, producing a signature (a mass spectrum) for each detected molecule (Figure 1, top).

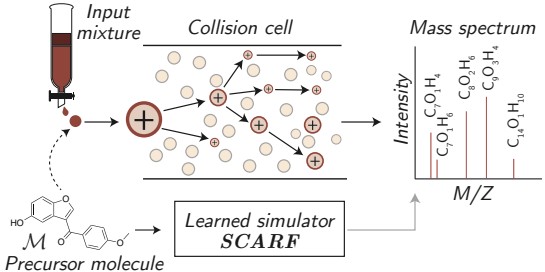

Figure 1: Tandem mass spectrometers measure fragmentation patterns of molecules, resulting in characteristic peaks that are indicative of their structure. SCARF simulates these fragmentation patterns *in silico*.

37th Conference on Neural Information Processing Systems (NeurIPS 2023).

Computationally predicting mass spectra from molecules *in silico* (Figure 1, bottom) is thus a longstanding and important challenge. Not only does this assist practitioners in better understanding the fragmentation process, but it also enables the identification of molecules from newly observed spectra by comparing an observed spectrum to virtual spectra generated from a database of candidate molecules. While a large library of empirical mass spectra could theoretically serve the same purpose, the size of such libraries is limited by the slow and expensive process of acquiring pure chemical standards and measuring their spectra, motivating computational prediction.

We argue that there are three core, interrelated desiderata for a forward molecule-to-spectrum simulation model, or "spectrum predictor". An ideal spectrum predictor should be (i) *accurate*, being able to predict the exact set of fragment masses and intensities with a precision comparable to experimental measurements; (ii) *physically inspired*, to avoid making physically nonsensical ("invalid") suggestions and to provide interpretations of the chemical species responsible for each peak for the benefit of human expert chemists; and (iii) *fast*, such that it is computationally inexpensive to predict spectra for many (e.g., millions) hypothetical molecules.

Unfortunately, many existing spectrum predictors do not meet these criteria. Methods to date have tended to follow one of two approaches: (a) physically motivated fragmentation approaches or (b) molecule-to-vector (or "binned") approaches (Figure 2A-B). Fragmentation approaches (e.g., [2, 17, 40, 52]; Figure 2A) take an input molecule and suggest bonds that may break, creating fragments that are scored by ML algorithms or curated rulesets. While interpretable, these methods are often slow and restrictive; certain mass spectrum peaks are generated by complex chemical rearrangements within the collision cell that cannot be approximated by bond breaking alone. That is, the bonds in observed fragments are not a subset of those in the original molecule [9, 11]. On the other hand, binned prediction approaches (e.g., [50, 55, 58]; Figure 2B) are less physically grounded, using neural networks to directly learn a mapping from molecules to vectors representing discretized versions of the spectra. These methods, while fast, lack interpretability and due to discretization have a mass precision lower than that of most modern spectrometers, limiting their accuracy.

We propose to address the shortcomings of previous work by predicting mass spectra from molecules at the level of molecular formulae (e.g., $C_xN_yO_zH_w...$) and introduce a new method, Subformulae Classification for Autoregressively Reconstructing Fragmentations (SCARF) to do so. Because the molecular formula for each input molecule is known, each subformula in the predicted set of peaks is constrained to contain a subset of the atoms in the original formula. Our primary contributions are:

- posing mass spectrum prediction as a two step process: first generating the set of molecular formulae for the fragments, then associating these formulae with intensities;

- overcoming the combinatorial subformula option space by learning to generate formula prefix trees;

- demonstrating the empirical benefit of SCARF in predicting experimental mass spectra quickly and accurately using two separate datasets, providing a benchmark for future work.

## 2 Background

We provide a short introduction of tandem mass spectrometry suitable for a general machine learning audience, detail previous approaches to modeling this process as they relate to our proposed approach SCARF, and explain how such tools can be utilized to discover molecules from new spectra. We refer interested readers to [25] for further details on the physical process of mass spectrometry.

### 2.1 Tandem mass spectrometry

Tandem mass spectrometers (MS/MS) measure fragmentation patterns of molecules in a multi-stage process. The input to the process is a solution containing a *precursor* molecule, $\mathcal{M} \in \mathcal{X}$, associated with a molecular formula, $\mathcal{F}$, defining the counts of each element present; for instance $\mathcal{F} = C_{16}O_4H_{12}$ for the precursor molecule shown in Figure 1. The precursor molecule is first ionized (i.e., made charged), often by bonding or associating with an *adduct* (e.g., a proton, $H^+$) present in the solution. The charged product is then measured by a mass analyzer (MS1), where its mass-to-charge ratio ($m/z$) is measured.

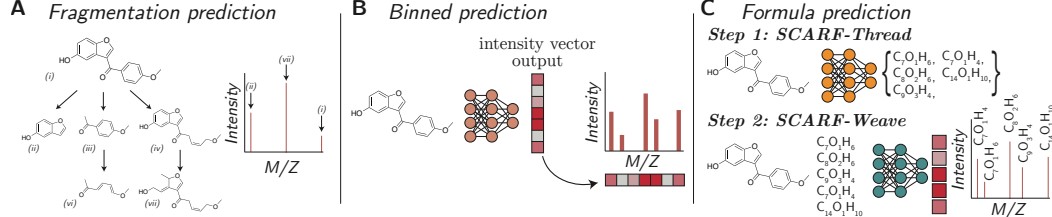

Figure 2: Overview of various approaches to spectrum prediction. **A.** Fragmentation prediction approaches use heuristics and scoring rules to break down the molecule into fragments and their associated intensities. **B.** Binned prediction approaches discretize the possible mass-to-charge values and predict intensities for each possible bin. **C.** Formula prediction approaches predict spectra as sets of molecular formulae and intensities. Our model SCARF utilizes a two stage approach, first by predicting the product formulae present (constrained by the precursor formula), which defines the x-axis locations of the peaks, before secondly assigning intensities to these formulae (defining the peaks' y-axis values).

This precursor ion is then filtered into a *collision cell*. Here, through interactions with an inert gas, the precursor ion is broken down into a set of one or more *product ions*, each of which is associated with a new chemical formula; for example, one might be $f^1 = C_7OH_4$ for the process shown in Figure 1. Finally, this set of product ions is measured by a second mass analyzer (MS2), along with the set of their intensities, $y^i \in \mathbb{R}^+$ (i.e., their relative frequencies over several repetitions of this process), creating for each ion what is referred to as a *peak*. The collection of all peaks makes up a molecule's *mass spectrum*, and is commonly represented as a plot of intensities versus $m/z$ (Figure 1, right).

## 2.2    Predicting mass spectra from molecules (spectrum predictors)

**Fragmentation prediction.**    A complex but physically grounded strategy is to model the bond breakage processes occurring in the collision cell (Figure 2A). Examples include MetFrag [52], MAGMa [40], and CFM-ID [2], which recursively fragment molecules (either bond or atom removals) to generate fragment predictions. These methods combine expert rules and local scoring methods to enumerate molecular fragmentation trees to predict spectra. CFM-ID [2] learns subsequent fragmentation transition probabilities between fragments with an expectation maximization algorithm to determine intensities at each fragment. Rule-based methods and full tree enumeration reduce the flexibility of these approaches, and along with the inherent ambiguity in the fragmentation process, limit this strategy's overall accuracy and speed.

**Binned prediction.**    An increasingly popular and straightforward approach to spectra prediction is to map molecules to discretized 1D mass spectra from either molecular fingerprint [50] or graph inputs [55, 58] (Figure 2B). Specifically, these methods divide the $m/z$ axis into fixed-width "bins" and predict an aggregate statistic of the peaks found in each bin (such as their maximum or summed intensity). While more flexible and end-to-end than fragmentation-based approaches, these methods do not impose the same physical constraints or shared information across fragments, making them less interpretable and susceptible to making invalid predictions. Further, discretizing the input spectrum inherently restricts the precision of such models compared to exact-mass predictions.

**Formula prediction.**    We introduce the strategy of predicting spectra at the level of molecular formulae, an intermediate between binned and fragmentation prediction (Figure 2C). Simultaneous to our work, two groups have separately explored formula prediction strategies [34, 59]. However, to generate plausible subformulae candidates, they either generate a fixed vocabulary of formulae [34] or restrict their model to molecules under 48 atoms for exhaustive enumeration [59], which is smaller than many compounds of interest. We overcome the combinatorial problem of formula generation using prefix trees, allowing our method to scale and eliminating the need for large, fixed vocabularies.

## 2.3 Mass spectrum libraries

One important use of spectrum predictors is in building large *in silico* libraries of molecule spectra to augment the small size of existing, experimentally derived databases (on the order of $10^4$) which are expensive to curate. These spectra libraries are then leveraged downstream in different ways, for example for training molecular property predictors directly from mass spectra [46]. Another common application of spectra libraries is to infer an unknown molecule's structure from a newly observed spectra – a particularly hard problem, with only 13% of spectra measured from clinical samples identifiable using current elucidation tools [6]. In this problem, spectra libraries are used as part of a process called *retrieval*: The newly observed spectra is compared with the existing spectra in the library using a fixed or learned spectral distance function, such as cosine distance [5, 24], and the molecules associated with the closest spectra are returned as possible matches. In practice, the retrieval process is constrained to choosing among *isomers* (i.e., molecules with the same molecular formula, and therefore molecular weight, but with different bond configurations) due to the high resolution of modern mass spectrometers (i.e., absolute errors on the order of $10^{-4}$ to $10^{-3}$ $m/z$ for MS1 measurements) [13, 33, 53].

Given the varied use cases of spectra libraries, we focus on evaluating spectrum predictors in terms of both (a) their prediction accuracies (§4.2), using metrics such as "cosine similarity", and (b) their use in generating virtual spectral libraries to assist with retrieval (§4.3).

## 3   Model

Here, we describe our model, SCARF, for predicting mass spectra from precursor molecules via first predicting subformulae of the precursor molecule, referred to as *product formulae*. Building upon the notation introduced in the previous section, we continue to denote precursor molecules[1] as $\mathcal{M} \in \mathcal{X}$, and their associated formula vector as $\boldsymbol{\mathcal{F}} \in \mathbb{N}_0^e$, defining at each position, $j \in \{1, \ldots, e\}$, the count of each possible chemical element present, $\mathcal{F}_j$ (with zero indicating none of that chemical element is present). Likewise, we define the set of $n$ product formulae as $\{\boldsymbol{f}^i\}_{i=1}^n$, and associate with each an intensity, $y^i$. Note that the mass[2] corresponding to a given formula (and, as such, the x-axis location of the peak on a mass spectrum) is determined deterministically from the counts of each elements present.

At a high level, SCARF generates mass spectra through the composition of two learned functions:

$$\left\{ (\boldsymbol{f}^i, y^i) \right\}_{i=1}^n = g_\theta^{\texttt{Weave}}\Big( g_\theta^{\texttt{Thread}}\left( \mathcal{M} \right), \mathcal{M} \Big), \tag{1}$$

first mapping from the original molecule to a set of product formulae, $g_\theta^{\texttt{Thread}} : \mathcal{M} \mapsto \{\boldsymbol{f}^i\}_{i=1}^n$, and then mapping from this set of formulae (and the original molecule) to the respective intensities, $g_\theta^{\texttt{Weave}} : (\{\boldsymbol{f}^i\}_{i=1}^n, \mathcal{M}) \mapsto \left\{ (\boldsymbol{f}^i, y^i) \right\}_{i=1}^n$. The particularities of both functions are described in detail below. The specific architectures and hyperparameters used are deferred to the appendix; model code can be found at https://github.com/samgoldman97/ms-pred.

### 3.1   SCARF-Thread : Generating product formulae via generating prefix trees

SCARF-Thread is tasked to learn a mapping to the set of product formulae, $\{\boldsymbol{f}^i\}_{i=1}^n$, given the original molecule. Naively, one might try to define this model autoregressively, predicting the set formula by formula, chemical element by chemical element. However, such an approach soon runs into a number of problems as (i) the predictions are not invariant to set and ordering permutations; (ii) the time complexity of prediction would scale poorly, being proportional to both the number of elements and number of product formulae (i.e., $\mathcal{O}(e \times n)$); and (iii) the predictions would likely contain duplicates.

---

[1]We model and discuss uncharged molecules and formulae, despite mass spectrometry measuring the masses of adduct *ions*. In practice, we reduce all molecules to uncharged candidates by simply shifting all the spectra weights by the $m/z$ of their respective adducts, which we assume to be equal to the (known) adduct of the parent molecule.

[2]We assume singly charged adducts (as is common practice, [2]), such that masses and mass-to-charge ratios are interchangeable.

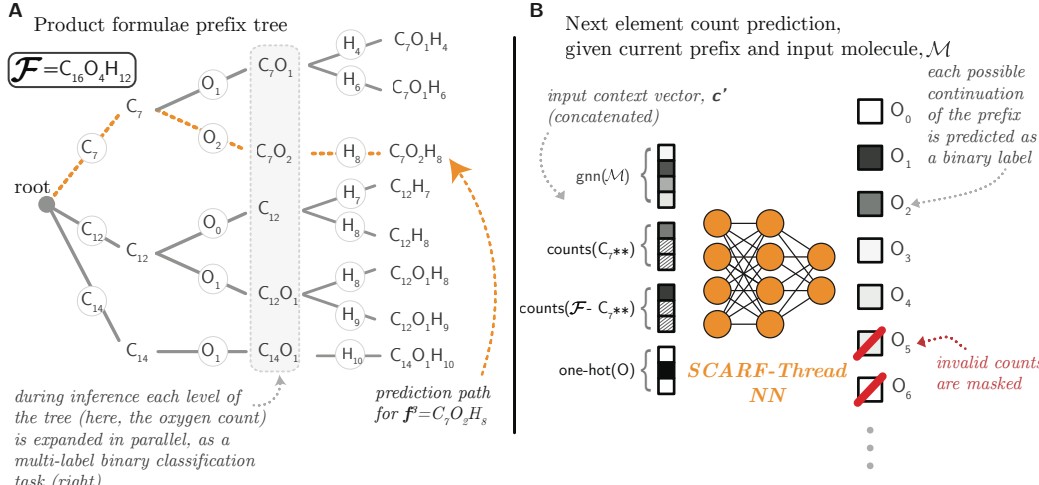

Figure 3: Illustration of the SCARF-Thread architecture. **A.** The formulae of the product fragments can be represented using a prefix tree. SCARF-Thread predicts this tree for new molecules at test time. It does so by expanding each node at a given depth in parallel, treating the counts of subsequent elements as dependent only on the counts of elements predicted so far (i.e., the prefix) and the original molecular structure. **B.** The SCARF-Thread predictive task at the $C_7$ node from the prefix tree diagram shown in A. Here the network takes as input (i) an embedding of the overall molecule; (ii) a vector representing the counts of each element in the prefix so far (counts yet to be predicted are represented using a special token), (iii) the difference of the counts predicted so far from the precursor molecule, and (iv) a one-hot representation of the element for which the counts are currently being predicted. The network predicts which counts are valid next nodes in the prefix tree (where counts that are greater than those in the original precursor molecular formula are automatically masked out as invalid). See also Alg. A.1.

We therefore take a different approach using the insight that the set of all product formulae can be compactly represented as a prefix tree (Figure 3A). In this tree, edges at a given depth represent valid counts of a particular chemical element, which are often identical across multiple product formulae (shown in the circles). By following each path from the root node to the different leaf nodes, we can reconstruct each product formula (as the orange dashed path does for a single product formula).

We thus propose SCARF-Thread as an autoregressive generator to define a probability distribution over such a prefix tree (Alg. A.1). We assume that each product formula is a subset of the precursor formula, meaning that the precursor formula sets an upper bound on the maximum number of each element[3]. At each node in the tree (corresponding to a prefix $f'_{<j}$), we pose the prediction of the set of child nodes (corresponding to the set of valid counts of the subsequent element) as a multi-label binary classification problem (Figure 3B). Concretely, we use a neural network module for this task, giving it as input a context vector representing the node being expanded:

$$c' = [\text{gnn}(\mathcal{M}), \text{counts}(f'_{<j}), \text{counts}(\mathcal{F} - f'_{<j}), \text{one-hot}(j)], \qquad (2)$$

where $\text{gnn}(\cdot)$ specifies a neural encoding of the molecular graph (§A.5.2), $\text{counts}(\cdot)$ specifies a count-based encoding of the associated prefix (§A.5.3), and $\text{one-hot}(\cdot)$ specifies a one-hot encoding of the node's depth (or equivalently, which element the predicted count is for). In our experiments, we use a fixed ordering of the chemical elements (§A.2), but optimizing or even learning the tree construction order could be carried out [45].

**Formulae as differences.** Following Wei et al. [50], we find it helpful to not only parameterize product formulae in terms of their element counts, but also in terms of the elements that they have lost,

---

[3]While it is possible for fragments to fuse together, potentially taking the count of a chemical element over the number in the original precursor formula, we postpone the extension to modeling such rare events to future work.

i.e., their *difference* from the precursor formula. On the input side, this is already covered by including in the context vector a count-based embedding of the prefix formula minus the product formula (counts($\mathcal{F} - \boldsymbol{f}'_{<j}$)). However, on the output side this is achieved by combining the probabilities of a "forward" and a "difference" network:

$$p(f'_j = a | \boldsymbol{f}'_{<j}, \mathcal{M}) = \boldsymbol{\alpha}_a \boldsymbol{\sigma} \left( \mathsf{MLP}^F(\boldsymbol{c}') \right)_a + (\mathbf{1} - \boldsymbol{\alpha})_a \boldsymbol{\sigma} \left( \mathsf{MLP}^D(\boldsymbol{c}') \right)_{\mathcal{F}_j - a}, \tag{3}$$

where $\mathsf{MLP}^F(\cdot)$ and $\mathsf{MLP}^D(\cdot)$ specify multi-layer perceptrons (MLPs) for predicting the probability of observing a count of $a$ and a loss of $\mathcal{F}_j - a$ atoms respectively; $\boldsymbol{\alpha}$ is a variable (output from a third, unshown network) deciding how to weight these predictions; and $\boldsymbol{\sigma}(\cdot)$ is the element-wise sigmoid function.

### 3.2 `SCARF-Weave`: Predicting intensities given product formulae

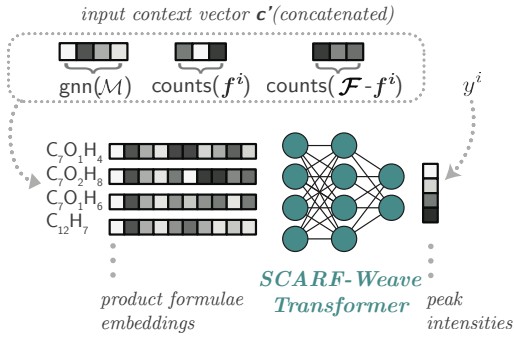

Figure 4: The `SCARF-Weave` network, which takes in the product formulae (e.g., predicted by `SCARF-Thread`) and predicts their intensities. We use a Set Transformer architecture [29], such that our model takes in the details of the other product formulae present when predicting intensities.

Given the product formulae outputs from `SCARF-Thread`, `SCARF-Weave` predicts corresponding intensities at each formula. This is a set-to-set problem, well suited for any equivariant set2set architecture [56, §3.1]. In our experiments, we use a Set Transformer [29, 44], which enables the model to consider all the formulae present in the mass spectrum (and their possible interactions) when predicting final intensities.

We choose to represent formula in the set similarly to the context vectors used in `SCARF-Thread`. For each input, we concatenate a vector embedding of the initial molecular graph with count-based embeddings of the product formula and its difference from the precursor formula (Figure 4). We again defer the particularities of the embedding functions to the Appendix (§A.5).

### 3.3 Training and inference

Provided with a dataset of molecules and formula-labeled mass spectra, we could train the two components of `SCARF` separately. However, in practice we find it beneficial to first train `SCARF-Thread` and then train `SCARF-Weave` on its outputs so that the distribution the latter model sees is the same at training and prediction time. `SCARF-Weave` is trained using a cosine loss (§A.5.5), as this most closely resembles the "retrieval" setting (§ 4.3).

`SCARF-Thread` is trained using the binary cross entropy losses associated with the multi-label classification tasks at each non-leaf node in the prefix tree. We use teacher forcing, i.e., we train on each level of the tree in parallel by conditioning on the ground-truth set of prefixes at each stage. In our experiments, when generating the set of product formulae from this model we always pick the top 300. Empirically, we find that this provides better performance than picking a variable number based on a likelihood threshold.

## 4 Experiments

We evaluate `SCARF` on spectra prediction (§ 4.2) and molecule identification in a retrieval task (§ 4.3).

### 4.1 Dataset

We train and validate `SCARF` on two libraries: a gold standard commercial tandem mass spectrometry dataset, `NIST20` [35], as well as a more heterogeneous public dataset, `NPLIB1`, extracted from the

Table 1: Model coverage (higher better) of true peak formulae as determined by `MAGMa` at various max formula cutoffs for the `NIST20` and `NPLIB1` datasets. Best result for each column is in bold. Results are computed for a single test set; all re-trained models (i.e., Autoregressive and SCARF variants) are averaged across three random seeds.

| Dataset | NIST20 | | | | NPLIB1 | | | |
|---|---|---|---|---|---|---|---|---|
| Coverage @ | 10 | 30 | 300 | 1000 | 10 | 30 | 300 | 1000 |
| Random | 0.009 | 0.026 | 0.232 | 0.532 | 0.004 | 0.014 | 0.126 | 0.336 |
| Frequency | 0.173 | 0.275 | 0.659 | 0.830 | 0.090 | 0.151 | 0.466 | 0.688 |
| CFM-ID | 0.197 | 0.282 | – | – | **0.170** | 0.267 | – | – |
| Autoregressive | 0.204 | 0.262 | 0.309 | 0.317 | 0.072 | 0.082 | 0.095 | 0.099 |
| SCARF-Thread-D | 0.248 | 0.425 | 0.839 | 0.941 | 0.158 | 0.284 | 0.681 | 0.856 |
| SCARF-Thread-F | 0.249 | 0.476 | 0.855 | 0.943 | 0.155 | 0.306 | 0.708 | 0.859 |
| SCARF-Thread | **0.308** | **0.552** | **0.907** | **0.968** | 0.164 | **0.309** | **0.724** | **0.879** |

GNPS database [48] by Dührkop et al. [14] and subsequently processed by Goldman et al. [19]. We prepare both datasets by extracting and preprocessing spectra, as well as filtering to compounds that (a) are under 1,500 Da (i.e., typically under 100 heavy atoms), (b) only contain predefined elements, and (c) are only charged with common positive-mode adduct types (§4.1).

Overall, `NIST20` contains 35,129 total spectra with 24,403 unique structures, and 12,975 unique molecular formulae; `NPLIB1` contains 10,709 spectra, 8,553 unique structures and 5,433 unique molecular formulae. Both datasets are evaluated using a structure-disjoint 90%/10% train/test split with 10% of training data held out for validation, such that all compounds in the test set are not seen in the train and validation sets.

**Annotating spectra.** We emphasize that `SCARF` can be trained with any product formula annotations, which can be labeled [35] or inferred with varied computational strategies [13]. Herein, we utilize the `MAGMa` algorithm [40]. In brief, for a given molecule-spectrum pair in the training dataset, the molecule is combinatorially fragmented at each atom up to a depth of 3 breakages to create sub-fragments. This creates a bank of possible molecular formulae, and each peak in the spectrum is assigned to its nearest possible formula within a mass difference of 20 parts-per-million.

## 4.2 Spectra prediction

**Predicting product formulae (`SCARF-Thread`).** `SCARF-Thread` is trained and used to reconstruct prefix trees and evaluated by its ability to recover the ground truth product formula set. The set of generated product formulae is rank-ordered by the probability of each product formula and filtered to the top $k$ predicted product formulae. The fraction of ground truth formulae (22.29 peaks on average in `NIST20`) contained in the top k set is computed as *coverage*.

We compare coverage achieved by `SCARF-Thread` to several baselines: (i) CFM-ID [2], a fragmentation based approach (§ A.3.1); (ii) a random baseline that samples product formulae from a uniform distribution; (iii) a frequency baseline, which ranks product formulae by the frequency the product formula candidate (or product formula difference) appears in the training set; (iv) an LSTM autoregressive neural network baseline (§ A.3.2) that is trained to predict molecular formula vectors in sequence from highest to lowest intensity; and two model ablations, (v) `SCARF-Thread`-D and (vi) `SCARF-Thread`-F, which only make uni-directional elements difference or forward predictions of element counts respectively (i.e., $\alpha$ in Equation 3 is fixed to **0** for (v) and **1** for (vi)).

In general, `SCARF-Thread` starkly outperforms all baselines tested (Table 1). By generating 300 peaks, `SCARF-Thread` is able to cover on average $91\%$ and $72\%$ of the true formulae in the ground truth test set for `NIST20` and `NPLIB1` respectively. Our difference- and forward-only directional prediction ablations demonstrate the benefits of modeling both the atom counts for each element and the differences in counts from the original molecule.

**Predicting mass spectra.** We next evaluate the strength of `SCARF-Weave` for intensity prediction on the same test dataset. We compare against five baselines: a fragmentation-based approach, `CFM-ID`

Table 2: Spectra prediction in terms of cosine similarity, coverage (proportion of ground-truth peaks that are covered by the top 100 non-zero predictions), validity (the fraction of predicted peaks for which a chemically plausible explanation is possible), and time. Best value in each column is typeset in bold (higher is better for all metrics but time). Results are averaged across 3 random seeds on a single data split for all retrainable models (i.e,. not CFM-ID).

| Dataset | NIST20 | | | NPLIB1 | | | |
|---|---|---|---|---|---|---|---|
| Metric | Cosine sim. | Coverage | Valid | Cosine sim. | Coverage | Valid | Time (s) |
| CFM-ID | 0.412 | 0.278 | **1.000** | 0.377 | 0.235 | **1.000** | 1114.7 |
| 3DMolMS | 0.510 | 0.734 | 0.945 | 0.394 | 0.507 | 0.919 | **3.5** |
| FixedVocab | 0.704 | 0.788 | 0.997 | **0.568** | **0.563** | 0.998 | 5.5 |
| NEIMS (FFN) | 0.617 | 0.746 | 0.948 | 0.491 | 0.524 | 0.949 | 3.9 |
| NEIMS (GNN) | 0.694 | 0.780 | 0.947 | 0.521 | 0.547 | 0.943 | 4.9 |
| SCARF | **0.726** | **0.807** | **1.000** | 0.536 | 0.552 | **1.000** | 21.1 |

[2]; two NEIMS binned prediction models ([50]; §A.3.3), using either feed forward network modules (FFNs), as in the original work, or graph neural network modules (GNNs) as in SCARF-Weave and described by Zhu et al. [58]; a retrained variant of 3DMolMS ([23]; §A.3.4), a binned spectrum predictor that utilizes a point cloud neural network over a single molecular conformer input generated by RDKit [39]; and FixedVocab, a formula prediction model that predicts intensities at a fixed library of formulae and formulae differences inspired by GRAFF-MS ([34]; §A.3.5).

To enable fair comparison across models, we predict test spectra at 15k bins (0.1 bin resolution between 0 and 1500) with a maximum of 100 peaks for each predicted molecule. With the exception of CFM-ID, all models are hyperparameter optimized (§A.5.6), retrained completely, and conditioned on the same covariate inputs as SCARF; such steps lead to large performance boosts to the prior NEIMS method in particular. We evaluate the quality of our predictions based upon four core criteria reflecting our original desiderata of accuracy, physical-sensibleness, and speed:

1. *Cosine sim.*: Cosine similarity between the ground truth and predicted spectra, indicating spectrum prediction accuracy.

2. *Coverage*: The fraction of ground truth spectrum peaks covered by the predicted spectrum.

3. *Valid*: The fraction of predicted peaks that can be explained by a subformula (that obeys basic ring-double bond equivalent heuristics [37]) of the predicted molecule.

4. *Time (s)*: The wall time it takes (using a single CPU and no batched calculations) to load the model and predict spectra for 100 randomly selected molecules.

SCARF is more accurate than all other approaches on NIST20, improving cosine similarity over a GNN binned prediction approach by over 0.02 points in NIST20 and 0.01 in NPLIB1 (Table 2). Further, our method is more physically grounded insofar as all predicted peaks are guaranteed to be valid subformulae, unlike the unconstrained binned approaches, where nearly 5% of peak predictions cannot be explained by a valid molecular formula. Importantly, SCARF still operates 2 orders of magnitude faster than CFM-ID (Table 2).

The heterogeneity, reduced dataset size, and increased average molecular weight (Figure A3) of NPLIB1 leads to substantially worse absolute performance across all models. Interestingly, in this setting, the FixedVocab approach [34] performs better, perhaps because the strict priors of formula constraints are more helpful with fewer and more challenging training examples. We further stratify results by molecule size in Figure A2, showing that all models are generally more accurate on smaller compounds. We additionally validate that cosine similarity is not merely measuring a model's ability to predict the parent mass peak by computing a modified cosine similarity with the original molecule's mass masked (Table A7).

### 4.3 Retrieval

A key application for forward spectrum prediction is to use predicted spectra to determine the most plausible molecular structure assignment. We posit forward spectrum prediction models should be

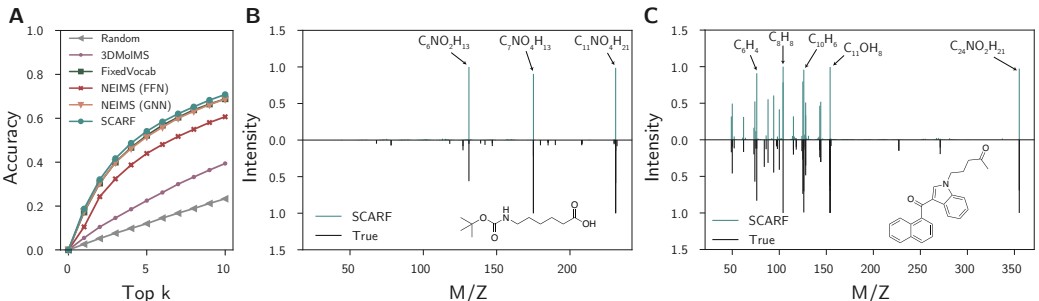

Figure 5: SCARF enables more accurate retrieval of ground truth molecules within the NIST20 dataset. **A.** Average retrieval accuracy of SCARF at various top k thresholds. Retrieval is conducted on the same test split, and retrieval accuracy is averaged across models trained for three separate random seeds. **B-C.** Example spectrum predictions made by SCARF (top) compared to the ground truth spectrum (bottom). Up to 5 predicted peaks with the highest intensity are annotated with their molecular formula explanation as predicted by SCARF. The full molecule is shown inset. Further examples are in the Appendix (Figure A1).

particularly helpful in differentiating structurally similar molecules and design a retrieval task to showcase such potential. For each test set molecule, we extract 49 potential "decoy" options based upon the most structurally similar *isomers* (i.e., compounds with the same precursor formula) within PubChem [27] as judged by Tanimoto similarity using Morgan fingerprints. While retrieval could be conducted on the entirety of PubChem or other similarly large molecular databases, we believe this subset retrieval setting is more practical and better mirrors a real-world setting (see §A.4 for justification). We predict the spectra for all molecules and rank them according to their similarity to the ground truth spectrum, computing the *accuracy* for retrieval. Herein, we specifically emphasize models and retrieval on the NIST20 dataset, as it is a much larger and higher quality dataset.

SCARF reaches a top 1 and top 5 retrieval accuracy in this task of 18.7% and 54.1% respectively, representing an improvement over the methods with the second best top 1 accuracy of 17.5% (NEIMS (GNN)) and top 5 accuracy of 52.2% (Fixed Vocab) (Figure 5A). We highlight two example predictions from SCARF (Figure 5B-C), with additional randomly sampled test set predictions shown in Figure A.1. We repeat similar experiments within NPLIB1, but find that cosine similarity performance is uncorrelated with relative ranking performance; feed forward fingerprint based approaches are better at retrieval, despite relatively weak cosine similarity (§A.1). FixedVocab [34] performs especially well on NPLIB1, again likely due to the helpful biases imparted by constraining the formula vocabulary.

This result underscores previous observations regarding how database and model biases can skew retrieval results under certain settings [22]. That is, models may be more or less robust for certain classes of molecules, so the composition of these classes in the retrieval library may affect the retrieval accuracy accordingly. The observed discrepancy between cosine similarity and retrieval performance can further be explained by the dataset shift required for computing retrieval accuracy; cosine similarity is evaluated on "in-domain" data, whereas retrieval relies also on accuracy on unlabeled data that may be "out-of-domain."

## 5 Related work

**Forward vs backward models.** Computational tools to identify mass spectra are often divided into two categories: forward and backward models. Forward models, i.e., spectrum predictors, such as SCARF or the methods discussed in Section 2, operate in the causal direction and try to predict the spectrum given the molecule. Backward models start from the spectrum and predict features or even full molecule structures. Early backward models used heuristics, expert rules, and even neural networks [8, 10, 42]. Such approaches have more recently been augmented with kernel methods and more modern, deep representation learning techniques [12, 16, 19, 47]. These models are complementary to spectrum predictors.

**Mass spectra for proteomics.** Although this paper has focused on small molecules, similar trends of deep representation learning for mass spectra are also emerging in the adjacent field of proteomics [54], with Shouman et al. [41] recently proposing a benchmark challenge in this domain. While small molecule and protein spectra are similar, proteomics spectra tend to be more easily predicted as fragments are often formed at peptide bonds. We believe adapting SCARF to this task would be an interesting direction for future work.

**Neural set generation.** Our work is also related to methods for modeling sets and multisets. SCARF-Thread generates a set as output, which has been studied elsewhere in the context of n-gram generation [45], object detection [32, 57], and point cloud generation [28]. The product formulae sets we generate, however, are different to those considered in these other task; in our setting, each member of the set (i.e., individual product formula) represents a multiset of atom types (i.e., multiple carbons, multiple hydrogens, etc.) and is constrained physically by the precursor formula.

## 6   Conclusion

In this paper we introduced SCARF, an approach utilizing prefix tree data structures to efficiently decode mass spectra from molecules. By first predicting product formulae and then assigning these formulae intensities, we are able to combine the advantages of previous neural and fragment based approaches, providing fast and physically grounded predictions. We show how these resulting predictions are both more accurate in predicting experimentally-observed spectra and yield improvements in identifying a molecule's structure from its respective mass spectrum, as tested on a widely used dataset.

In term of limitations, our model is data dependent, as indicated by the relative performance across the NIST20 and NPLIB1 datasets. SCARF is also highly reliant upon the quality of product formula label assignment. The current commercial status of mass spectrometry training data poses a barrier to entry, and identifying high quality public domain data is critical for future studies. A key contribution of this work is to retrain and optimize the hyperparameters of competing methods on a publicly available dataset under equivalent conditions to allow for future extensions. Directly training on a ranking-based loss or learning a model specific spectra distance function may be one way to improve upon our model's performance in the retrieval setting, and we outline additional potential ideas more explicitly in §A.6.

Future directions will involve further developing SCARF for real world use cases such as unknown metabolite elucidation in clinical samples. Specific directions will include more carefully modeling covariates (e.g., collision energies and MS/MS instrument types), grounding product formulae in molecular graph substructures, and utilizing such models to augment inverse spectrum-to-molecule annotation tools.

## Acknowledgements

We thank members of the Coley Research Group, as well as Michael Murphy, for helpful discussions and comments. S.G. thanks the MIT-Takeda Program for financial support. J.B., J.X., and C.W.C. thank the Machine Learning for Pharmaceutical Discovery and Synthesis consortium for additional support.

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

# A Appendix

## A.1 Extended results

We benchmark models in terms of retrieval accuracy as described (§ 4.3) for both the `NIST20` and `NPLIB1` datasets (Table A1, A2). We recreate retrieval experiments using the full PubChem retrieval library in Table A3. We reproduce main text results with standard error values included in Tables A4, A5, and A6. We showcase additional spectra predictions from our model trained on `NIST20` in Figure A1.

Table A1: `NIST20` spectra prediction retrieval top k accuracy for different values of k. All values represent the mean across three separate random seeds $\pm$ the standard error of the mean for a single test set.

| top k | 1 | 2 | 3 | 4 | 5 | 8 | 10 |
|---|---|---|---|---|---|---|---|
| Random | $0.026 \pm 0.000$ | $0.052 \pm 0.001$ | $0.076 \pm 0.001$ | $0.098 \pm 0.001$ | $0.120 \pm 0.000$ | $0.189 \pm 0.001$ | $0.233 \pm 0.002$ |
| 3DMolMS | $0.055 \pm 0.002$ | $0.105 \pm 0.000$ | $0.146 \pm 0.002$ | $0.185 \pm 0.004$ | $0.225 \pm 0.004$ | $0.332 \pm 0.003$ | $0.394 \pm 0.004$ |
| FixedVocab | $0.172 \pm 0.002$ | $0.304 \pm 0.002$ | $0.399 \pm 0.001$ | $0.466 \pm 0.004$ | $0.522 \pm 0.006$ | $0.638 \pm 0.005$ | $0.688 \pm 0.003$ |
| NEIMS (FFN) | $0.105 \pm 0.002$ | $0.243 \pm 0.006$ | $0.324 \pm 0.006$ | $0.387 \pm 0.006$ | $0.440 \pm 0.007$ | $0.549 \pm 0.005$ | $0.607 \pm 0.002$ |
| NEIMS (GNN) | $0.175 \pm 0.003$ | $0.305 \pm 0.001$ | $0.398 \pm 0.001$ | $0.462 \pm 0.002$ | $0.515 \pm 0.003$ | $0.632 \pm 0.003$ | $0.687 \pm 0.003$ |
| SCARF | $\mathbf{0.187 \pm 0.004}$ | $\mathbf{0.321 \pm 0.006}$ | $\mathbf{0.417 \pm 0.004}$ | $\mathbf{0.486 \pm 0.004}$ | $\mathbf{0.541 \pm 0.005}$ | $\mathbf{0.652 \pm 0.004}$ | $\mathbf{0.708 \pm 0.005}$ |

Table A2: `NPLIB1` spectra prediction retrieval top k accuracy for different values of k. All values represent the mean across three separate random seeds $\pm$ the standard error of the mean for a single test set.

| top k | 1 | 2 | 3 | 4 | 5 | 8 | 10 |
|---|---|---|---|---|---|---|---|
| Random | $0.033 \pm 0.001$ | $0.061 \pm 0.005$ | $0.092 \pm 0.003$ | $0.118 \pm 0.003$ | $0.141 \pm 0.006$ | $0.216 \pm 0.006$ | $0.258 \pm 0.006$ |
| 3DMolMS | $0.087 \pm 0.001$ | $0.159 \pm 0.010$ | $0.218 \pm 0.004$ | $0.268 \pm 0.006$ | $0.317 \pm 0.006$ | $0.427 \pm 0.008$ | $0.488 \pm 0.005$ |
| FixedVocab | $0.193 \pm 0.003$ | $\mathbf{0.314 \pm 0.004}$ | $\mathbf{0.390 \pm 0.003}$ | $\mathbf{0.448 \pm 0.005}$ | $\mathbf{0.492 \pm 0.001}$ | $\mathbf{0.587 \pm 0.005}$ | $0.635 \pm 0.006$ |
| NEIMS (FFN) | $\mathbf{0.195 \pm 0.003}$ | $0.313 \pm 0.002$ | $0.388 \pm 0.003$ | $0.447 \pm 0.006$ | $0.488 \pm 0.002$ | $0.585 \pm 0.007$ | $0.624 \pm 0.010$ |
| NEIMS (GNN) | $0.174 \pm 0.007$ | $0.285 \pm 0.004$ | $0.362 \pm 0.002$ | $0.422 \pm 0.001$ | $0.471 \pm 0.002$ | $0.586 \pm 0.007$ | $\mathbf{0.640 \pm 0.005}$ |
| SCARF | $0.135 \pm 0.007$ | $0.242 \pm 0.001$ | $0.320 \pm 0.001$ | $0.389 \pm 0.004$ | $0.444 \pm 0.002$ | $0.569 \pm 0.001$ | $0.630 \pm 0.008$ |

Table A3: Retrieval accuracy on `NIST20` for a single 500 molecule subset of the test set using a library of 49 decoys or all decoys contained in PubChem ("None"). Results were repeated for 3 random training seeds of the model and are shown $\pm$ the standard error of the mean. The top value is shown in bold.

| PubChem limit | 50 | | | None | | |
|---|---|---|---|---|---|---|
| Top k | 1 | 2 | 3 | 1 | 2 | 3 |
| FixedVocab | $0.168 \pm 0.003$ | $0.308 \pm 0.003$ | $0.410 \pm 0.005$ | $0.145 \pm 0.004$ | $\mathbf{0.258 \pm 0.004}$ | $0.323 \pm 0.001$ |
| NEIMS (FFN) | $0.102 \pm 0.002$ | $0.237 \pm 0.003$ | $0.315 \pm 0.004$ | $0.087 \pm 0.003$ | $0.183 \pm 0.008$ | $0.236 \pm 0.007$ |
| NEIMS (GNN) | $0.169 \pm 0.003$ | $0.300 \pm 0.004$ | $0.402 \pm 0.005$ | $0.138 \pm 0.004$ | $0.239 \pm 0.005$ | $0.312 \pm 0.008$ |
| SCARF | $\mathbf{0.204 \pm 0.009}$ | $\mathbf{0.326 \pm 0.005}$ | $\mathbf{0.432 \pm 0.009}$ | $\mathbf{0.167 \pm 0.003}$ | $\mathbf{0.258 \pm 0.001}$ | $\mathbf{0.336 \pm 0.003}$ |

## A.2 Dataset preparation

`NIST20` [35] is prepared by extracting all positive-mode experimental spectra collected in higher-energy collision-induced dissociation (HCD) mode (i.e., collected on Orbitrap mass spectrometers). Spectra are filtered, so that we keep only those for which the associated molecule (M) has (i) a mass under 1,500 Da, (ii) contains only elements from a predefined set (i.e,. "C", "N", "P", "O", "S", "Si", "I", "H", "Cl", "F", "Br", "B", "Se", "Fe", "Co", "As", "Na", "K"), and (iii) is charged with common adduct types (i.e., "[M+H]+", "[M+Na]+", "[M+K]+", "[M-H2O+]+", "[M+NH3+H]+", and "[M-2H2O+H]+"). Because non-standard empirical spectra databases [48] often do not include the measured collision energies, we pool all collision energies for each compound-adduct pairing to create a single spectrum. We refer the reader to Young et al. [55] for detailed instructions for purchasing and extracting the `NIST20` dataset.

All spectrum intensities are square-root transformed to provide higher weighting to lower intensity peaks, normalized to a maximum intensity of 1 (i.e., through dividing by the maximum intensity),

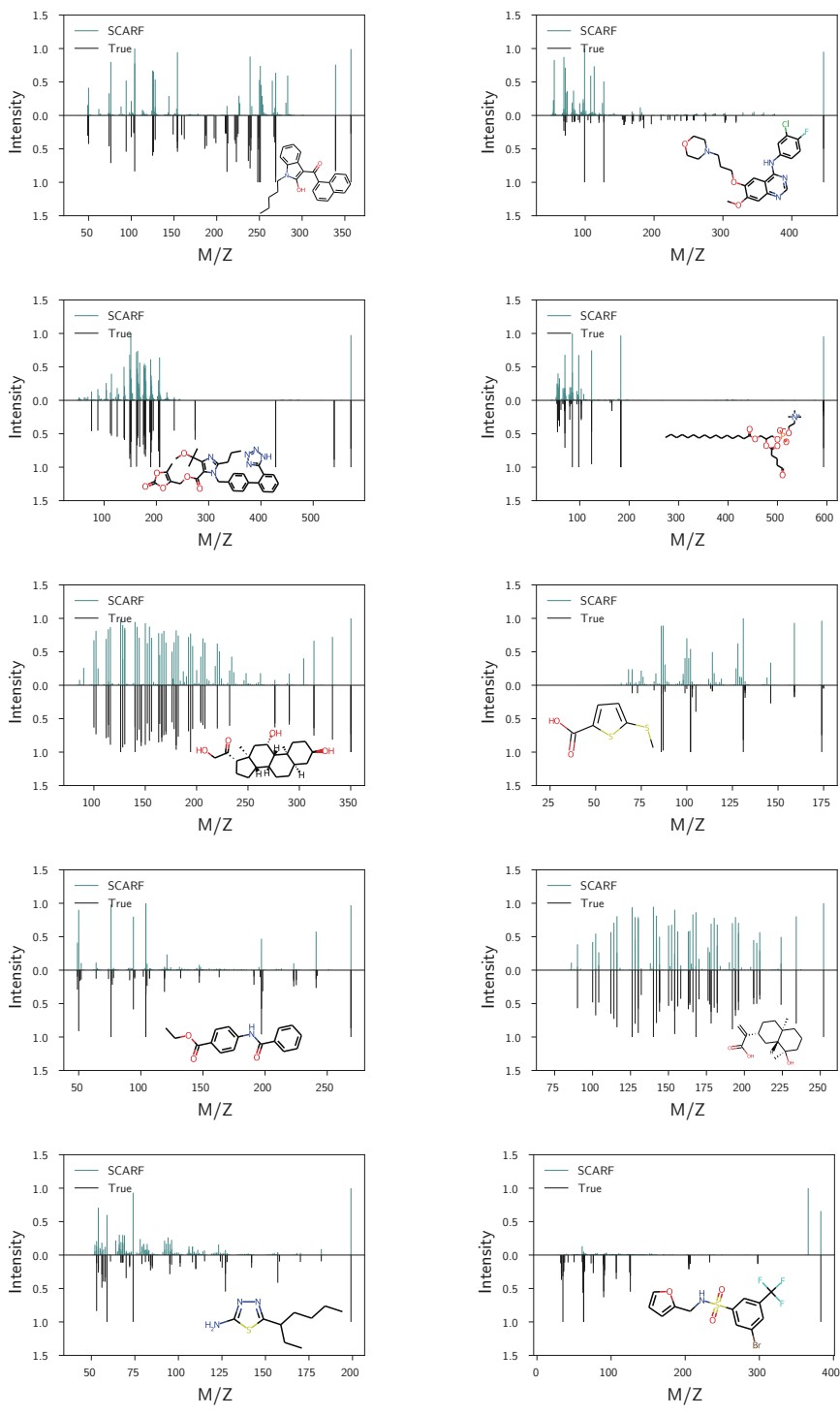

Figure A1: Example spectra predictions from the NIST20 dataset for 10 randomly selected test molecules. The ground truth spectra are shown underneath in black, with predictions above in teal. Molecules are shown inset.

Table A4: Model coverage of true peak formulae as determined by `MAGMa` at various max formula cutoffs for the `NPLIB1` dataset. Results are calculated for a single held out test split, shown $\pm$ the standard error of the mean across three random seeds for all models that were retrained. The best value (i.e., highest) is typeset in bold for each column.

| Coverage @ | 10 | 30 | 300 | 1000 |
|---|---|---|---|---|
| Random | 0.004 | 0.014 | 0.126 | 0.336 |
| Frequency | 0.090 | 0.151 | 0.466 | 0.688 |
| CFM-ID | **0.170** | 0.267 | – | – |
| Autoregressive | $0.072 \pm 0.001$ | $0.082 \pm 0.002$ | $0.095 \pm 0.001$ | $0.099 \pm 0.000$ |
| SCARF-D | $0.158 \pm 0.001$ | $0.284 \pm 0.003$ | $0.681 \pm 0.002$ | $0.856 \pm 0.002$ |
| SCARF-F | $0.155 \pm 0.002$ | $0.306 \pm 0.003$ | $0.708 \pm 0.003$ | $0.859 \pm 0.001$ |
| SCARF | $0.164 \pm 0.009$ | $\mathbf{0.309 \pm 0.014}$ | $\mathbf{0.724 \pm 0.013}$ | $\mathbf{0.879 \pm 0.004}$ |

Table A5: Model coverage of true peak formulae as determined by `MAGMa` at various max formula cutoffs for the `NIST20` dataset. Results are calculated for a single held out test split, shown $\pm$ the standard error of the mean across three random seeds for all models that were retrained. The best value (i.e., highest) is typeset in bold for each column.

| Coverage @ | 10 | 30 | 300 | 1000 |
|---|---|---|---|---|
| Random | 0.009 | 0.026 | 0.232 | 0.532 |
| Frequency | 0.173 | 0.275 | 0.659 | 0.830 |
| CFM-ID | 0.197 | 0.282 | – | – |
| Autoregressive | $0.204 \pm 0.001$ | $0.262 \pm 0.002$ | $0.309 \pm 0.005$ | $0.317 \pm 0.006$ |
| SCARF-D | $0.248 \pm 0.001$ | $0.425 \pm 0.002$ | $0.839 \pm 0.002$ | $0.941 \pm 0.001$ |
| SCARF-F | $0.249 \pm 0.001$ | $0.476 \pm 0.002$ | $0.855 \pm 0.000$ | $0.943 \pm 0.001$ |
| SCARF | $\mathbf{0.308 \pm 0.002}$ | $\mathbf{0.552 \pm 0.001}$ | $\mathbf{0.907 \pm 0.002}$ | $\mathbf{0.968 \pm 0.001}$ |

filtered to exclude any noise peaks with normalized intensity under 0.003, and subsetted to only the top 50 highest intensity peaks. All peaks are mass-shifted by the weight of the parent adduct (i.e,. if the spectrum is "[M+H]+", the weight of a proton is subtracted from each child peak).

### A.2.1 Product formulae assignments

Because the precursor ion and adduct species are known for the training dataset, we subtract the precursor adduct mass from every peak in the training set, and attempt to annotate each peak with a plausible product formula (i.e., a subset of the true precursor formula).

We opt to constrain the training product formulae to be subsets of contiguous heavy atoms of the parent molecule as derived with the `MAGMa` algorithm [40].

We note two important limitations of these heuristics. First, by using molecular substructures to annotate product formulae, our model is less prone to correctly identifying complex rearrangements. Second, it is also possible for adduct switching to occur. Namely, if the precursor ion has a sodium adduct ("[M+Na]+"), some of the product formulae may actually switch and acquire a hydrogen adduct instead. We assume no adduct switching in our formulation, instead focusing on the novelty of the prefix tree decoding approach, as these represent data labeling challenges, rather than modeling challenges.

In addition, any predictive models of product formulae distributions will more closely predict spectra that would be produced on instrumentation similar to the training sets utilized [9, 11]. Given this, we encourage users of such models to treat these predictions as putative, rather than experimentally valid.

Table A6: Spectra prediction in terms of cosine similarity, coverage (proportion of ground-truth peaks that are covered by the top 100 non-zero predictions), validity (the fraction of predicted peaks for which a chemically plausible explanation is possible), and time. Best value in each column is typeset in bold (higher is better for all metrics but time). Values are shown $\pm$ the standard error of the mean computed across three random seeds on a single test set for all models that could be retrained (i.e., not CFM-ID).

| Dataset | NIST20 | | | NPLIB1 | | | |
| --- | --- | --- | --- | --- | --- | --- | --- |
| | Cosine sim. | Coverage | Valid | Cosine sim. | Coverage | Valid | Time (s) |
| CFM-ID | $0.412 \pm 0.000$ | $0.278 \pm 0.000$ | $\mathbf{1.00 \pm 0.000}$ | $0.377 \pm 0.000$ | $0.235 \pm 0.000$ | $\mathbf{1.00 \pm 0.000}$ | 1114.7 |
| 3DMolMS | $0.510 \pm 0.000$ | $0.734 \pm 0.001$ | $0.94 \pm 0.001$ | $0.394 \pm 0.002$ | $0.507 \pm 0.001$ | $0.92 \pm 0.000$ | $\mathbf{3.5}$ |
| FixedVocab | $0.704 \pm 0.000$ | $0.788 \pm 0.000$ | $\mathbf{1.00 \pm 0.000}$ | $\mathbf{0.568 \pm 0.002}$ | $\mathbf{0.563 \pm 0.001}$ | $\mathbf{1.00 \pm 0.000}$ | 5.5 |
| NEIMS (FFN) | $0.617 \pm 0.000$ | $0.746 \pm 0.001$ | $0.95 \pm 0.001$ | $0.491 \pm 0.002$ | $0.524 \pm 0.001$ | $0.95 \pm 0.000$ | 3.9 |
| NEIMS (GNN) | $0.694 \pm 0.000$ | $0.780 \pm 0.000$ | $0.95 \pm 0.001$ | $0.521 \pm 0.002$ | $0.547 \pm 0.003$ | $0.94 \pm 0.001$ | 4.9 |
| SCARF | $\mathbf{0.726 \pm 0.001}$ | $\mathbf{0.807 \pm 0.000}$ | $\mathbf{1.00 \pm 0.000}$ | $0.536 \pm 0.007$ | $0.552 \pm 0.008$ | $\mathbf{1.00 \pm 0.000}$ | 21.1 |

Table A7: Spectra prediction accuracy comparing inclusion (Cosine sim.) and exclusion (Cosine sim. (no MS1)) of the precursor mass. For all compounds, the peak at the mass of the input compound is masked in the prediction and ground truth to compute Cosine sim. (no MS1). All results represent an average on a single test set across three random seeds.

| Dataset | NIST20 | | NPLIB1 | |
| --- | --- | --- | --- | --- |
| | Cosine sim. | Cosine sim. (no MS1) | Cosine sim. | Cosine sim. (no MS1) |
| CFM-ID | 0.412 | 0.289 | 0.377 | 0.326 |
| 3DMolMS | 0.510 | 0.517 | 0.394 | 0.390 |
| FixedVocab | 0.704 | 0.637 | **0.568** | **0.505** |
| NEIMS (FFN) | 0.617 | 0.557 | 0.491 | 0.454 |
| NEIMS (GNN) | 0.694 | 0.620 | 0.521 | 0.477 |
| SCARF | **0.726** | **0.663** | 0.536 | 0.466 |

### A.2.2 Dataset statistics

To probe the composition of our two primary datasets, we investigate both the molecular weight and chemical classes contained in the NPLIB1 and NIST20 datasets. We find that the average molecular weight is higher for NPLIB1 (Figure A3A), consistent with the increased complexity of natural product molecules. We additionally compute chemical classes of the compounds using NPClassifier [26] to identify the types of compounds present in both datasets (Figure A3B-C). NPLIB1 is enriched for steroids, coumarins, and various complex alkaloid natural products. On the other hand, NIST20 is enriched for small peptides, nicotinic acid alkaloids, and pseudoalkaloids, among others. While these descriptions are helpful to identify the dataset composition, chemical compound classification is itself a learned classification and should be interpreted cautiously.

Table A8: NIST20 retrieval accuracy averaged across three random seeds of model training stratified by the weight of the target molecule.

| Dataset | NIST20 | | | | | | |
| --- | --- | --- | --- | --- | --- | --- | --- |
| Molecular weight | 0 - 200 | 200 - 300 | 300 - 400 | 400 - 500 | 500 - 600 | 600 - 700 | 700 - 2000 |
| Num. compounds | 624 | 1358 | 900 | 387 | 129 | 56 | 89 |
| Random | 0.024 | 0.021 | 0.027 | 0.025 | 0.031 | 0.048 | 0.101 |
| 3DMolMS | 0.039 | 0.041 | 0.057 | 0.077 | 0.070 | 0.125 | **0.199** |
| FixedVocab | 0.143 | 0.184 | **0.168** | 0.172 | **0.196** | **0.220** | 0.161 |
| NEIMS (FFN) | 0.110 | 0.122 | 0.092 | 0.078 | 0.111 | 0.083 | 0.064 |
| NEIMS (GNN) | 0.155 | 0.192 | 0.164 | **0.182** | 0.147 | 0.214 | 0.161 |
| SCARF | **0.191** | **0.211** | 0.165 | 0.163 | 0.168 | 0.161 | 0.169 |

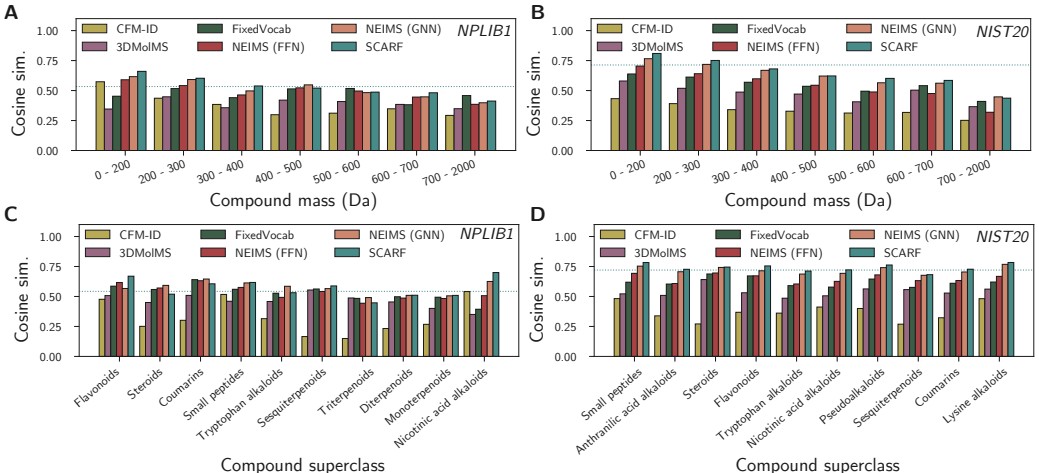

Figure A2: Cosine similarity of predicted spectra is stratified across molecular weight for both NPLIB1 (**A**) and NIST20 (**B**). We further stratify results across putative chemical classes of input molecules using NPClassifier [26] for both NPLIB1 (**C**) and NIST20 (**D**). The dotted line indicates the average predictive cosine similarity of SCARF across all examples and averaged over three random splits.

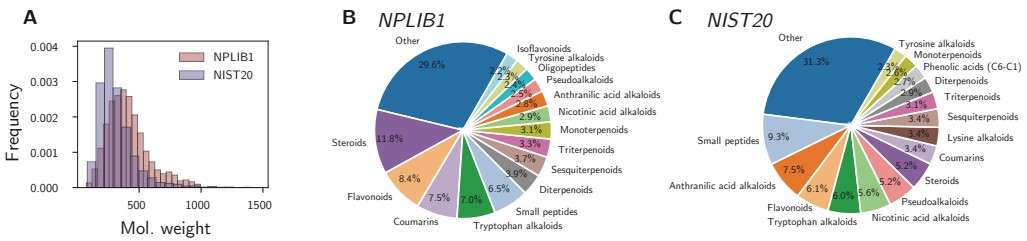

Figure A3: Spectra dataset molecule characterizations. **A.** Distribution of the molecular weight of compounds across NPLIB1 and NIST20. **B-C.** Chemical classes contained in NPLIB1 (B) and NIST20 (C) with the top 15 classes shown and all others grouped in 'Other'. Chemical classes are computed using NPClassifier [26].

## A.3 Baselines

We further describe select baseline models.

### A.3.1 CFM-ID baseline

CFM-ID [2] is a long standing and important approach to fragmentation prediction. Because CFM-ID is fit using a time intensive EM training approach on an analogous dataset, we utilize the pretrained Docker implementation provided by the authors in line with [34]. CFM-ID has two options for predicting molecules in either positive or negative adduct mode with "H" adducts (i.e., "[M+H]+" or "[M-H]-"). To directly compare to our method, we predict spectra in positive mode and remove hydrogens from all predicted peaks, as all training peaks are shifted by removing their adducts.

CFM-ID also produces predictions at three collision energies (i.e., low , medium, or high fragmentation). Because we opt not to include these, we merge these predictions and re-normalize the result to a maximum of 1.

### A.3.2 Autoregressive baseline

When considering the task of generating spectrum formulae candidates, we compare `SCARF-Thread` to an autoregressive recurrent neural network baseline, which is based around a long short term memory (LSTM) module [21].

The LSTM generates formulae consecutively from a single concatenated encoding of the input molecule and input full formula. At each step in the recurrent process, a one-hot encoding of the previous predicted element count is concatenated to a one-hot encoding of the element type being predicted in the current step. By embedding this information into the network, we can avoid predicting element types that do not appear in the parent molecule's molecular formula. If the parent molecular formula has 5 element types, each autoregressively predicted formula requires generating only 5 element type counts; this eliminates the need for a stop condition between each formula. Formulae are generated autoregressively, from highest to lowest intensity. When predicting the counts of the next element type, we employ the same difference and forward count prediction strategy as used in `SCARF` for fair comparison. The model is trained with a cross entropy loss and full hyperparameters are listed in Table A10.

### A.3.3 NEIMS baseline

NEIMS [50] is a highly efficient binned spectrum prediction approach, originally developed for gas chromatography-mass spectra (GC-MS). To enable a fair comparison, we optimize its hyperparameters on our dataset and add higher resolution bins. Furthermore, we also train a graph neural network-based version "NEIMS (GNN)" (in addition to the network that more closely matches Wei et al. [50]'s original model and operates on the molecular fingerprint, "NEIMS (FFN)"). The adduct type is either concatenated to all atom features (for NEIMS (GNN)) or to the fingerprint vector (for NEIMS (FFN)).

### A.3.4 3DMolMS baseline

3DMolMS [23] is a binned spectrum prediction approach developed simultaneously to this work. Unlike the other binned NEIMS approach, 3DMolMS utilizes a point-based deep neural network model operating on the point cloud of an input molecule. To project a 2D molecule or SMILES string into 3D space, a single 3D conformer is first generated using RDKit [39]. After several 3D convolutions, the atom-wise representations are pooled, covariates corresponding to the settings of the machine and experiment are concatenated, and the result is projected into a fixed length binned spectrum.

To enable a fair comparison, we copy the 3DMolMS architecture into our modeling framework with minor tweaks to the network. Rather than use variable sized hidden layers, we fix hidden layer sizes to a single value across convolutions. In addition, we only use the covariate of the adduct type for consistency with our model, excluding collision energy and instrument type. We hyperparameter optimize the model independently. We find that the performance of this model is substantially lower than the NEIMS baseline, likely due to the additional use of the "difference" prediction module in the NEIMS approach that allows the network to predict intensities at both fragments and neutral losses.

### A.3.5 FixedVocab baseline

Concurrently to our work, Murphy et al. introduce an alternative formula prediction strategy for mass spectrum prediction, a model they term GRAFF-MS. Unlike `SCARF`, GRAFF-MS utilizes a fixed vocabulary of molecular formulae and molecular formula differences, predicting intensities at each such value without learning to encode the formulae. These formulae and formula differences are selected in a greedy fashion based upon their frequency in the training set.

Because no code was released for this approach at the time this work was conducted, we reimplement a variant of their approach that emphasizes the use of a fixed vocabulary of formulae and differences. For methodological consistency, we utilize equivalent formula annotations as used by `SCARF` (i.e., one annotation per peak), do not model collision energies or instrument types, and utilize the same graph encoder as `SCARF` for encoding each molecule. We treat the number of fragment and difference formulae as a tunable hyperparameter (which is optimized along with the rest of the hyperparameters – see §A.5.6). We mask all invalid formulae and differences and utilize a cosine similarity loss with the original spectrum to train the model. To convert predicted formula and difference intensities into

a binned spectrum, each formula-intensity pairing is projected into its respective binned position using a scatter max calculation. We note that because alternative adducts and isotopes are not labeled in our preprocessing step, we do not predict isotopic or adduct variants for each fragment.

Given the differences between codebases, it is possible that the performance of our reimplementation does not exactly match the original implementation, and we instead refer to it as "FixedVocab" rather than GRAFF-MS in table presentations. An earlier version of our work understated the FixedVocab model's performance due to an implementation decision along these lines (specifically, not including the "0" neutral loss as a predicted vocabulary entry). This has since been rectified, increasing the accuracy of the FixedVocab model.

### A.4    Retrieval subsets vs. PubChem

We restrict our retrieval experiments in §4.3 to only the top 49 decoys per test case for two reasons. First, from a practical perspective, running the forward model on every isomer match in PubChem (approx. thousands each, >1,000,000 for only 500 test cases) makes benchmarking across all considered models substantially more challenging both for this work and also future work. Second, we also believe that this top 50 challenge represents a more realistic setting. In practice, retrieval compound libraries will often be carefully crafted and designed to contain molecules similar to the unknown molecule rather than all possible isomers (using either prior knowledge or "backward" models such as CSI:FingerID [12] and MIST [19]). We conduct a side-by-side analysis on a small 500 molecule subset of the test set comparing the setting described above (with 49 decoys) to a setting with no limit on the number of decoys. The results are shown in Table A3, showing that SCARF still performs well in this setting.

### A.5    Model details

Here, we describe details of our model's training setup, architecture, and hyperparameters that were omitted from the main text. Definitive details can also be found in the code at `https://github.com/samgoldman97/ms-pred`.

### A.5.1    Training

We train each of our models on a single RTX A5000 NVIDIA GPU (CUDA Version 11.6), making use of the Torch Lightning [15] library to manage the training. SCARF-Thread and SCARF-Weave take on the order of 1.5 and 2.5 hours of wall time to train respectively.

### A.5.2    Molecule encoding

Within both SCARF-Thread and SCARF-Weave, a key component is an encoding of the molecular graph using a message passing graph neural network, $gnn(\mathcal{M})$. Such graph neural network models are now well described [3, 7, 20], so we will skip a detailed explanation of them here. In our experiments, we use gated graph sequence neural networks [30]. We made use of the implementation of this network in the DGL library [49] and use as atom features those shown in Table A9 (which are computed using RDKit [39] or DGL [49]).

Table A9: Graph neural network (GNN) atom features.

| Name | Description |
|------|-------------|
| Element type | one-hot encoding of the element type |
| Degree | one-hot encoding of number of bonds atom is associated with |
| Hybridization type | one-hot encoding of the hybridization (SP, SP2, SP3, SP3D, SP3D2) |
| Charge | one-hot encoding of atom's formal charge (from -2 to 3) |
| Ring-system | binary flag indicating whether atom is part of a ring |
| Atom mass | atom's mass as a `float` |
| Chiral tag | atom's chiral tag as one-hot encoding |
| Adduct type | one-hot encoding of the adduct ion |
| Random walk embed steps | positional encodings of the nodes computed using DGL |

### A.5.3 Molecular formulae representations

When forming representations of formulae (including formulae prefixes) we use a count-based encoder, $\text{counts}(\boldsymbol{f})$. This encoder takes in as input the counts of all individual elements in the formula (which also can be "undefined" for counts of elements not yet specified – indicated as '$*$' in Figure 3B) and returns a vector representation in $\mathbb{R}^d$. The encoder is based upon the Fourier feature mapping proposed by Tancik et al. [43], but using only $\sin$ basis functions (to reduce the number of parameters required by our networks). Tancik et al. [43] has shown that such features perform better than encoding integers directly; furthermore, compared to learned representations, using Fourier features enables us (at least in principle) to deal with counts at test time that have not been seen during training.

To be precise, each possible count, $v \in \mathbb{N}_0$, is encoded by our counts-based encoder into the vector:

$$\text{abs}\left(\left[\sin\left(\frac{2\pi v}{T_1}\right), \sin\left(\frac{2\pi v}{T_2}\right), \sin\left(\frac{2\pi v}{T_3}\right), \dots\right]\right),$$

where the periods ($T_1$, $T_2$, etc.) are set at increasing powers of two that enable us to discriminate all possible element counts given in the input, and $\text{abs}(\cdot)$ is the absolute value function such that we get positive embeddings. For the "undefined" count we learn a separate encoding of the same dimensionality.

### A.5.4 Further details of SCARF-Thread

Pseudo-code for the SCARF-Thread model is shown in Algorithm A.1. Note that the second for loop (on the line marked ‡) does not depend on previous iterations of the loop, so that in practice we perform this computation in parallel. At training time we use teacher forcing (§3.3), meaning the first for loop (marked †) is only run sequentially at inference time.

The function scarf-thread-net($\cdot$) represents the network shown in Figure 3B and generates the set of subsequent valid element counts given a prefix (i.e., the child nodes of a given prefix node). As discussed in the main text, we treat this as a multi-label binary classification task and predict the binary label for each possible count using forward and difference MLPs (Eq. 3). We fix a maximum possible element count (i.e., the number of possible classes in this classification problem), $N = 160$. We do not allow product formulae to have more of a given element than is present in the precursor formula, $\mathcal{F}$, and we achieve this by setting the probability of these classes to zero.

---

**Algorithm A.1:** Pseudo-code for SCARF-Thread, which generates prefix trees from a root node autoregressively, one level at a time.

---

**Data:** Input molecule, $\mathcal{M}$, with corresponding input formula, $\mathcal{F}$.
**Result:** Set of product formulae, $\rho_e = \{\boldsymbol{f}^i\}_{i=1}^n$.

1   $\boldsymbol{h}_{\mathcal{M}} \leftarrow \text{gnn}(\mathcal{M})$ ;        ▷ Form embedding of precursor molecule.
2   $\rho_0 \leftarrow \{*\}$ ;        ▷ Store the set of initial prefixes which is just the undefined formula, $*$.
3† **for** $j \in [1, \dots, e]$ **do**        ▷ Loop over all possible elements.
4     $\rho_j \leftarrow \{\,\}$ ;        ▷ Create the set of prefixes the next time around.
5     $\boldsymbol{h}_j \leftarrow \text{one-hot}(j)$ ;        ▷ Encoding of which element we are predicting the count of.
6‡     **for** $\boldsymbol{f}'_{<j} \in \rho_{j-1}$ **do (in parallel)**        ▷ Loop over all current prefixes.
7       $\boldsymbol{c}' = [\boldsymbol{h}_{\mathcal{M}}, \text{counts}(\boldsymbol{f}'_{<j}), \text{counts}(\mathcal{F} - \boldsymbol{f}'_{<j}), \boldsymbol{h}_j]$ ;        ▷ Create context vector, Eq. 2
8       $\{f_j^{i'}\}_{i'=1}^{n'} \leftarrow \text{scarf-thread-net}(\boldsymbol{c}', \mathcal{F})$ ;        ▷ Predict the set of valid next element counts under this prefix.
9       $\rho_j \leftarrow \rho_j \cup \text{create-new-prefixes}(\boldsymbol{f}'_{<j}, \{f_j^{i'}\}_{i'=1}^{n'})$ ;        ▷ Create new prefixes for the next element.
   **return** $\rho_e$

---

### A.5.5 Further details of SCARF-Weave

As discussed in the main text, SCARF-Weave is based off Lee et al. [29]'s Set Transformer. After forming the input encoding using the molecule and count-based encoder (§A.5.2 & §A.5.3), we further refine this embedding using an MLP (multi-layer perceptron) network. The output of this is passed into a series of $l_3$ Transformer [44] layers (§A.5.6 defines the exact number used in the experiments) with 8 attention heads each.

We use a cosine distance loss to train the parameters of `SCARF-Weave`. This loss is also used for the FFN and GNN baselines (Table 2). To ensure consistency with the baselines, we first project the output of `SCARF-Weave` into a binned histogram representation (§A.5.6 defines the number of bins used); for each bin we take the max intensity across all applicable formulae. Given a predicted binned spectra, $\hat{s}$, and the ground-truth binned spectra, $s$, the cosine distance is defined as the negative of the cosine similarity (computed using PyTorch's `torch.cosine_similarity` function [36]):

$$\text{cos-sim}\,(\hat{s}, s) = \frac{\hat{s} \cdot s}{\max\,(\|\hat{s}\|_2 \|s\|_2,\ \epsilon)}, \tag{A.1}$$

where $\epsilon = 1 \times 10^{-8}$ is used to ensure numerical stability.

### A.5.6 Hyperparameters

To enable fair comparison across models, hyperparameters were tuned for `SCARF`, the FFN binned prediction baseline, and the GNN binned prediction baseline. Parameters were tuned using RayTune [31] with Optuna [1] and an ASHAScheduler. Each model was allotted 50 different hyperoptimization trials for fitting. Models were hyperparameter optimized on a smaller $10,000$ spectra subset of `NIST20`. Parameters are detailed in Table A10.

### A.6 Limitations and future work

We outline several potential directions for future work to address limitations of this work.

1. *Improving the gold standard training annotation pre-processing.* Because `SCARF` is flexible in that it can match distributions of formula assignments, a key step to improving and building upon this approach is to develop more robust assignments of formula to training spectra. This includes adding complexity and removing potential assumptions, such as allowing annotations to account for rearrangement, elimination, or charge transfer. A second goal is to identify potentially low quality training spectra, such as ones that emerge from mixtures, and remove these from the inputs. Another potential way to handle such cases would be to model each spectrum peak as an *ensemble* of potential equivalent-mass formulae, which would be particularly helpful in relating `SCARF` to inverse models such as MIST [19] in which the structure of the molecule and formula identity of each peak cannot be known *a priori*.

2. *Incorporating other model covariates.* Incorporating collision energy features explicitly into the model, as well as negative-ion mode inputs, will increase its usability. This could be enabled by aggregating public data containing these annotations.

3. *Featurizing molecule inputs using different or more powerful molecular encoders.* Recent and simultaneous work to this used a pretrained graph encoder as part of a binned spectrum prediction approach, MassFormer [55]. It is possible to include more powerful molecule or formula encoders into `SCARF`.

4. *Consideration of interpretability by subgraph attribution and combination with ICEBERG.* Following this initial work, we developed a second model, ICEBERG [18], that uses a similar two step modeling approach, but instead encodes fragments, not formula. This increases accuracy and robustness, especially for retrieval, but substantially slows the model. In comparison to ICEBERG, `SCARF` still has several benefits including speed, the lack of required substructure labeling, and ability to capture potential skeletal rearrangements of molecules (i.e., discontinuities in structure that may not be possible to model by only breaking bonds). An open question and exciting opportunity in the future is to combine these two levels of abstraction and make formula-level predictions with graph-level attribution or featurization.

5. *Retrieval-specific loss functions to enhance retrieval performance.* A significant finding of this work was the noise associated with the retrieval task and lack of correlation with spectrum prediction performance as measured by cosine similarity. Future work may consider how to more directly define loss functions that reflect the task of retrieval.

Table A10: Model and baseline hyperparameters.

| Model | Parameter | Grid | Value |
|---|---|---|---|
| Autoregressive | learning rate | $[1e-4, 1e-3]$ | 0.0009 |
| | learning rate scheduler | - | StepDecay (5,000) |
| | learning rate decay | $[0.7, 1.0]$ | 0.85 |
| | dropout | $\{0.0, 0.1, 0.2, 0.3\}$ | 0.2 |
| | hidden size, $d$ | $\{128, 256, 512\}$ | 512 |
| | gnn layers | $[1, 6]$ | 1 |
| | rnn layers | $[1, 3]$ | 3 |
| | batch size | $\{8, 16, 32, 64\}$ | 64 |
| | weight decay | $\{0, 1e-6, 1e-7\}$ | $1e-6$ |
| | use differences (Eq.3) | $\{True, False\}$ | True |
| | conv type | - | GatedGraphConv |
| | random walk embed steps (Table A9) | $[0,20]$ | 20 |
| | graph pooling | $\{mean, attention\}$ | mean |
| NEIMS (FFN) | learning rate | $[1e-4, 1e-3]$ | 0.00087 |
| | learning rate scheduler | - | StepDecay (5,000) |
| | learning rate decay | $[0.7, 1.0]$ | 0.722 |
| | dropout | $\{0.0, 0.1, 0.2, 0.3\}$ | 0.0 |
| | hidden size, $d$ | $\{64, 128, 256, 512\}$ | 512 |
| | layers, $l$ | $\{1, 2, 3\}$ | 2 |
| | batch size | $\{16, 32, 64, 128\}$ | 128 |
| | weight decay | $\{0, 1e-6, 1e-7\}$ | 0 |
| | use differences (Eq.3) | $\{True, False\}$ | True |
| | num bins (§A.5.5) | - | $15,000$ |
| NEIMS (GNN) | learning rate | $[1e-4, 1e-3]$ | 0.00052 |
| | learning rate scheduler | - | StepDecay (5,000) |
| | learning rate decay | $[0.7, 1.0]$ | 0.767 |
| | dropout | $\{0.0, 0.1, 0.2, 0.3\}$ | 0.0 |
| | hidden size, $d$ | $\{64, 128, 256, 512\}$ | 512 |
| | layers, $l$ | $[1, 6]$ | 4 |
| | batch size | $\{16, 32, 64\}$ | 64 |
| | weight decay | $\{0, 1e-6, 1e-7\}$ | $1e-7$ |
| | use differences (Eq.3) | $\{True, False\}$ | True |
| | num bins (§A.5.5) | - | $15,000$ |
| | conv type | - | GatedGraphConv |
| | random walk embed steps (Table A9) | $[0,20]$ | 19 |
| | graph pooling | $\{mean, attention\}$ | mean |
| 3DMolMS | learning rate | $[1e-4, 1e-3]$ | 0.00074 |
| | learning rate scheduler | - | StepDecay (5,000) |
| | learning rate decay | $[0.7, 1.0]$ | 0.86 |
| | dropout | $\{0.0, 0.1, 0.2, 0.3\}$ | 0.3 |
| | hidden size, $d$ | $\{64, 128, 256, 512\}$ | 256 |
| | layers, $l$ | $[1, 6]$ | 2 |
| | top layers | $[1, 3]$ | 2 |
| | neighbors, $k$ | $[3, 6]$ | 5 |
| | batch size | $\{16, 32, 64\}$ | 16 |
| | weight decay | $\{0, 1e-6, 1e-7\}$ | $1e-6$ |
| | num bins (§A.5.5) | - | $15,000$ |
| FixedVocab | learning rate | $[1e-4, 1e-3]$ | 0.00018 |
| | learning rate scheduler | - | StepDecay (5,000) |
| | learning rate decay | $[0.7, 1.0]$ | 0.92 |
| | dropout | $\{0.0, 0.1, 0.2, 0.3\}$ | 0.3 |
| | hidden size, $d$ | $\{64, 128, 256, 512\}$ | 512 |
| | layers, $l$ | $[1, 6]$ | 6 |
| | batch size | $\{16, 32, 64\}$ | 64 |
| | weight decay | $\{0, 1e-6, 1e-7\}$ | $1e-6$ |
| | num bins (§A.5.5) | - | $15,000$ |
| | conv type | - | GatedGraphConv |
| | random walk embed steps (Table A9) | $[0,20]$ | 11 |
| | graph pooling | $\{mean, attention\}$ | mean |

| Model | Parameter | Grid | Value |
|---|---|---|---|
| | formula library size | $\{1000, 5000, 10000, 25000, 50000\}$ | 5000 |
| `SCARF-Thread` | learning rate | $[1e-4, 1e-3]$ | 0.000577 |
| | learning rate scheduler | - | StepDecay (5,000) |
| | learning rate decay | $[0.7, 1.0]$ | 0.894 |
| | dropout | $\{0.0, 0.1, 0.2, 0.3\}$ | 0.3 |
| | hidden size, $d$ | $\{128, 256, 512\}$ | 512 |
| | mlp layers, $l_1$ | $[1, 3]$ | 2 |
| | gnn layers, $l_2$ (§A.5.2) | $[1, 6]$ | 4 |
| | batch size | $\{8, 16, 32, 64\}$ | 16 |
| | weight decay | $\{0, 1e-6, 1e-7\}$ | $1e-6$ |
| | use differences (Eq.3) | $\{$True, False$\}$ | True |
| | conv type | - | GatedGraphConv |
| | random walk embed steps (Table A9) | $[0,20]$ | 20 |
| | graph pooling | $\{$mean, attention$\}$ | mean |
| `SCARF-Weave` | learning rate | $[1e-4, 1e-3]$ | 0.00031 |
| | learning rate scheduler | - | StepDecay (5,000) |
| | learning rate decay | $[0.7, 1.0]$ | 0.962 |
| | dropout | $\{0.0, 0.1, 0.2, 0.3\}$ | 0.2 |
| | hidden size, $d$ | $\{128, 256, 512\}$ | 512 |
| | mlp layers, $l_1$ (§A.5.5) | $[1, 3]$ | 2 |
| | gnn layers, $l_2$ (§A.5.2) | $[1, 6]$ | 3 |
| | transformer layers, $l_3$ (§A.5.5) | $[0, 3]$ | 2 |
| | batch size | $\{4, 8, 16, 32, 64\}$ | 32 |
| | weight decay | $\{0, 1e-6, 1e-7\}$ | 0 |
| | num bins (§A.5.5) | - | $15,000$ |
| | conv type | - | GatedGraphConv |
| | random walk embed steps (Table A9) | $[0,20]$ | 7 |
| | graph pooling | $\{$mean, attention$\}$ | attention |

