# OpenReview forum: "Prefix-Tree Decoding for Predicting Mass Spectra from Molecules"
_NeurIPS.cc/2023/Conference — NeurIPS 2023 spotlight_

### Official Review · Reviewer_8EJR · 2023-06-28

**Soundness:** 4 excellent
**Presentation:** 4 excellent
**Contribution:** 4 excellent
**Rating:** 8
**Confidence:** 4

**Summary:**

This paper proposes a novel method for predicting a tandem mass spectrum from a given molecular graph.  The key idea is to subdivide the problem into predicting fragmentation of the graph and then separately predicting intensities associated with each fragment.  Relative to other recent methods that have adopted this approach, this method uses a prefix tree to make the identification of subgraphs more efficient.  Experimental results show that the proposed method performs better than state-of-the-art methods.

**Strengths:**

The paper addresses an important problem.

The paper is extremely well written, with a nice intro to tandem mass spectrometry targeted to a NeurIPS audience.

The treatment of related work is well organized and clear.

The main idea is well supported by empirical results.


**Weaknesses:**

On lines 37-42, I wasn't super convinced that "physically-inspired" (which, incidentally, doesn't need to be hyphenated) belongs in this list of desiderata.  I think the main thing is that the method is fast and accurate.  It seems more like a hypothesis to say that in order to achieve speed and accuracy, a method should be physically inspired.  Perhaps the notion of "accurate" should be segregated into various dimensions: e.g., accurate in the sense that it doesn't just predict binned m/z values, or accurate in the sense that it doesn't predict extra peaks that can't be explained by the formula.  I would argue that if someone came along with a method that was very accurate and fast, there is no scenario in which an end user would prefer a slightly slower or slightly less accurate method just because it comes with a story about how it is physically plausible.

Figure 1A is the key idea, and I think it needs to be expanded.  As drawn, it's not clear how, e.g., you decide that the root should have three children (C7, C12 and C14).  Why not C1, C2, etc.?  The answer presumably has to do with the molecular graph, which is the primary input but is not shown here.  I would also like to hear a bit more about this prefix tree step in the text.  E.g., does it matter what order the atom types are introduced?   If so, how is the order chosen?

Related to the first point above, the claim on lines 100-102 that binning methods are less interpretable was hard for me to understand.  The reduced accuracy due to the lack of shared information across fragments is plausible, but that seems like an empirical claim that requires support.  Maybe just say that you hypothesize that this may lead to reduced accuracy. I also found it strange that you didn't explicitly call out the most obvious source of inaccuracy, which is that the binning means that the peak m/z locations are inherently inaccurate.

For the SCARF-Weave predictor, please indicate how large the sets can be come, both in principle and in practice.  I wonder whether this is a potential problem for the method.

When you present the various competing methods on lines 248-255, I think it would be helpful if you could introduce them in terms of the three groups from Section 2.2.

The closest competing methods are references [24] and [45].  In your experimental comparisons, why did you use a method based on MS-GRAFF ([24]) rather than just using MS-GRAFF itself?  And why didn't you use method [45] at all?

On a related note, if FixedVocab falls into the third category of methods, then why aren't its predictions 100% valid in Table 2?


**Questions:**

Is there a concrete sense in which binning methods are less interpretable?

How large are the sets handled by SCARF-Weave?

Why are MS-GRAFF and FormulaNet/SubsetNet not included in the empirical comparison?

---

> ### Author Rebuttal · Authors · 2023-08-08
>
> Thank you for your very constructive comments and suggestions to help improve the work!
>
> **W1:** “Physically inspired” should not necessarily be a desiderata
> **A**: We agree that the desiderata are somewhat interrelated and your point is well taken (and thanks — we have removed the hyphen!). We hypothesize that to achieve accuracy — particularly in low data or out of distribution domains — a method should be physically inspired (or at least pre-trained on a body of similar and physically relevant data), which may help to "parameterize out'' certain false positive errors. We had originally used "interpretable" as a term (removed to disambiguate from interpretability about how the networks are working) and are open to suggestions.
>
> **W2** Fig 1A can be clarified
> **A**: Assuming that you are referring to Figure 3A, we will make this Figure clearer to indicate that each element count number is predicted as a function of the input molecule and also reference pseudocode in the Supplementary Material (A.4.4). In answer to your specific questions:
> - This figure shows the full prefix tree (from the product formulae set). At training time only the correct nodes would be expanded (i.e., the three children you mention, because they appear in the ground truth spectrum). At inference time, the model would predict (as a multi-label binary classification problem) which nodes to expand (as such it will sometimes make mistakes).
> - The order of the atom types is fixed based on the order in which we define the vocabulary of elements, but it would be interesting to explore whether different orders would lead to better or worse performance.
>
> **W3**: The "most obvious source of inaccuracy" is missing for binned spectra.
> **A**: We attempted to point this out (line 54), and we will also add it to this later section! We will also clarify interpretability (see answer to Q1 below).
>
> **W4**: Is the SCARF-Weave predictor limited in how large sets can be?
> **A**: We were able to train models with up to 500 predicted formulae in reasonable time on a single GPU (<5 hours) given the modest size of the dataset (approx. 35k spectra) We anticipate this number could be pushed higher but would likely require switching the pairwise inter-formula attention to a sparse attention framework. Of course, this does increase the time for inference.
>
> **W5** Other formula-based methods are not included as baselines.
> **A**: We address this in our response to Q3 below.
>
> **W6**: FixedVocab does not have 100% validity.
> **A**: Our  “validity” metric checks if all peaks can be explained by a formula that passes the “ring double bond equivalent” heuristic for valence rules. Because FixedVocab also allows for the prediction of “loss” formulae, there are edge cases on which the parent molecule formula minus a “valid” loss prediction leads to an “invalid” fragment prediction. Such cases are very rare; any method that uses formula-based or structure-based representations should have roughly 100% validity.
>
> **W7**: The closest competing methods are references [24] and [45]
> **A**: See Q3 below.
>
> **Q1**: Is there a concrete sense in which binning methods are less interpretable?
> **A**: Binning methods provide no provenance for the peaks, or even if an individual bin is actually made up of many smaller peaks. SCARF (providing formulae) or fragmentation methods (providing substructures) provide further information, giving practitioners clues to interpret why a predicted spectrum looks the way it does (linking back to the original molecule's properties). These additional outputs can also be used to assess the reliability of the prediction.
>
> **Q2**: How large are the sets handled by SCARF-Weave?
> **A**: As described above (W4), SCARF-Weave sets were fixed at 300 candidate formulae, but we pushed them up to 500 during internal experiments. This could likely be increased further at the expense of speed but may require switching to sparse attention.
>
> **Q3**: Why are MS-GRAFF and FormulaNet/SubsetNet not included in the empirical comparison?
> **A**: We became aware of these methods (both published earlier this year), while finalizing our submission. In turn:
>
> _GRAFF-MS._ At the time of our submission, we were unable to obtain code or data for GRAFF-MS; the code has since been released approximately two weeks ago, at its presentation at the ICML conference (July 25, 2023). FixedVocab is our attempt at a reimplementation (from the details that existed in the GRAFF-MS preprint). To match our other models, we reimplemented it using 1) our own formula labeling scheme (rather than NIST’s internal tool that has isotopic peak labels); 2) the same GNN architecture as SCARF for consistency; and 3) the same cosine loss that we utilized to train SCARF.  We renamed it as “FixedVocab” to avoid misrepresenting the original work, but the core idea of the two models should be essentially the same.
>
> _FormulaNet/SubsetNet._ We did not compare to FormulaNet/SubsetNet as it is trained using a maximum of 48 atoms (because it requires complete enumeration of formula candidates), a key limitation preventing its use on our tested datasets (NIST20 has molecules with up to 102 heavy atoms). It is trained for EI spectra acquired in combination with gas phase chromatography and more useful for more “volatile” molecule classes where this size limitation is less problematic.

---

> > ### Comment · Reviewer_8EJR · 2023-08-10
> >
> > I have read the other reviews and the authors' rebuttal. I found the rebuttal clear and convincing, particularly with respect to the choice of competing methods.

---

### Official Review · Reviewer_188s · 2023-07-06

**Soundness:** 4 excellent
**Presentation:** 4 excellent
**Contribution:** 3 good
**Rating:** 7
**Confidence:** 5

**Summary:**

The manuscript presents a method called SCARF for the prediction of tandem mass spectra from molecular structures. This allows for building up large in-silico libraries that, together with measured spectra, can be used in spectral library search workflows for metabolite identification. From a general machine learning perspective, the method is an interesting approach for the problem of predicting sets of multisets (in this case: sets of molecular formulas). The author note that autoregression is not well suited for this task, as it is not ordering invariant. Instead, they suggest an approach that is quite similar to the way molecular formulas are decomposed from masses: they step-by-step build a prefix-tree where each node represents a subset of the molecular formula space with certain elements are fixed to specific abundancies. The child nodes are then possible abundancies for the next chemical element, restricting the formula space further until the the leafs of the tree describe only a single molecule formula. The resulting set of molecular formulas is then transformed into a spectrum, by utilizing a second transformer based network for intensity prediction.

**Strengths:**

- it is the first paper that predicts high resolution spectra without involving combinatorial fragmentation

- use of prefix tree for the prediction of molecular formulas (or sets of multisets) is very innovative

- use of random Fourier features for molecular formula embedding is innovative and seem to work much better than the common approaches of encoding formulas as frequency vectors or one-hot vectors/dictionaries

- evaluation on the spectrum prediction part is done very well. In particular, certain methods are retrained (or even re-implemented) and parameter-optimized to ensure fair comparison

- manuscript is clearly written, the method is described in detail with all hyper parameters given in the supplement

- source code is available

**Weaknesses:**

- there is already a successor method (ICEBERG) by the same authors for the same task. This is not a problem in general, as both methods have very different strategies to solve the problem. I still think it would have strengthen the manuscript if pros and cons of both methods and their synergistic potential is discussed

- the manuscripts list three applications for the predicted spectra: molecule identification, learning about fragmentation, and training machine learning methods on spectra. However, in contrast to ICEBERG the method presented here cannot give any insights about the fragmentation process (as it is not a fragmenter). I also doubt that synthetic data (predicted machine learning) is well suited as training data for other machine learning methods. This is some kind of self-training or self-supervised learning which is a difficult and challenging problem on its own. Thus, the only convincing application for the spectrum prediction task is the molecule identification. Beside spectrum prediction, there are other methods that allow for molecule identification (including methods from the same authors such as MIST). Although the evaluation on metabolite identification contains many methods for spectrum prediction, it does not contain any of these alternative approaches.

- the observation that identification rate is decoupled from database retrieval rate is concerning, as this is the main application for spectrum prediction

- although all methods perform worse on the NPLIB1 data, it is strange that NEIMS(FFN) and SCARF perform so very different on the  retrieval task. It would be interesting to find out why the fingerprint based method performs and the dictionary based method both perform so well here but not in NIST.

- besides its application in metabolomics,  the presented approach might be interesting for other areas with sets of multisets as target variable. Unfortunately, the baseline method (autoregression) is not evaluated against the more sophisticated prefix tree decoding method

**Questions:**

- does the cosine similarity calculation takes the precursor ion into account? For many spectra, the precursor can have high intensity (and, thus, heavily effect cosine similarity) while it does not provide any new information. If so, I would suggest to recalculate cosine similarity and leave out the precursor
- I do not understand why the spectrum retrieval task is performed on a subset of 49 isomers (although the most similar ones) instead of just all isomers

**Limitations:**

The main limitation of the method is in my opinion its limited performance on the metabolite identification task. This and other limitations are discussed properly in the manuscript.

---

> ### Author Rebuttal · Authors · 2023-08-08
>
> Thank you for your very fair comments and feedback.
>
> **W1**: Putting SCARF in context with other recent methods
> **A:** Thanks for your suggestion. We will add a section to the Appendix explaining how SCARF has more flexibility as it does not require structural fragment labeling, higher speed, cosine similarity, but struggles on retrieval. We agree that SCARF and other models (including even older fragmentation-based approaches) have synergies and think developing ways to combine them (e.g., mixtures of experts)  would be an interesting future direction.
>
> **W2.1**: Of the three stated applications of such a model (data augmentation, interpretation, and molecule identification), molecule identification is the only one that is relevant.
> **A:** This is an interesting point; thank you for raising it. While we agree one could argue that molecule identification is currently probably the most promising aspect of these models, we want to push back a little here about it being the "only convincing application".
>
> For example, the MIST manuscript [14; p.44]  reported that data augmentation (via synthetic data) is surprisingly very beneficial, which follows a theme similar to how AlphaFold2 (Jumper et al., p.587) was able to generate increased performance with knowledge distillation methods. We strongly suspect that better / synthetic augmentation (perhaps generated from SCARF) will be critical to improving backwards models, which we hope future studies will prove.
>
> **W2.2**: Molecule identification is not compared against other methods for molecule identifications such as MIST
> **A** The comparison between other molecular identification methods such as fingerprint prediction or contrastive learning is an interesting, open question. For the purposes of this work, which we intend to contribute a new forward predictive method, we see that as somewhat out of scope (also absent in all existing work on spectrum prediction [36,44,24]). One interesting facet of spectra predictors compared to backwards models is that they work in the causal direction and so may perform differently when tested on out of distribution data (Kilbertus et al., 2018); we hope a future benchmark of all molecular identification methods also includes such tasks.
>
> > Kilbertus, N., Parascandolo, G. and Schölkopf, B. (2018) ‘Generalization in anti-causal learning’,: http://arxiv.org/abs/1812.00524.
>
> **W3-4**: The decoupling between retrieval and cosine accuracy is perplexing
> **A**: We agree with this and had attempted to highlight this in our paper. We believe this discrepancy results from cosine similarity accuracy being computed on labeled, “in-domain” data, whereas retrieval relies also on accuracy on unlabeled data. We will revisit this in the paper accordingly to make it more clear.
>
> Part of this somewhat ties into your previous question (about application). If ultimately, one decides one wants to use a spectra predictor purely for retrieval it may make sense to consider directly training on a ranking-based loss or learning a model specific spectra distance function, but we leave such follow ups to future work.
>
> **W5**: Autoregressive model baseline for formula generation.
> **A**: We agree this could help clarify our point. We have implemented such an approach where an autoregressive language model successively predicts each formula in order of highest to lowest intensities. We find that, in preliminary experiments on NIST20, this model has top 10, 30, and 300 coverage of 0.214, 0.276, and 0.324 vs. SCARF-Thread’s respective coverage of 0.316, 0.559,  and 0.911.  The limitation of this baseline is likely due to the long range dependencies involved and tendency to stop generating unique formulae after a certain position (i.e., for a molecule of CHNOP elements, generating 300 potential product formulae would require 1,500 tokens). This is aside from the other issues we mention (lines 147-151).
>
> **Q1**: Cosine similarity should be recomputed without precursor ions
> **A**: We have conducted this additional analysis (see Author Response Table 2), masking out the mass of the full parent chemical formula in both the predicted and ground truth spectrum. Under these new conditions, SCARF is still the most performant on NIST20, but fails to outperform FixedVocab on NPLIB1 (consistent with poor retrieval on this dataset).
>
> Through this analysis, we uncovered that our implementation of FixedVocab prevented it from predicting the parent chemical formula and have updated it (included in the table). We will update all results for FixedVocab in the full manuscript accordingly; thank you for helping us to notice this. We note that we have also recomputed top 1 retrieval accuracy on NIST20 with this new method and it is 0.177 (vs. 0.165 in Appendix Table A1); SCARF is still superior with a Top 1 accuracy of 0.184.
>
> **Q2**: Spectrum retrieval should be performed against the entirety of PubChem
> **A**: To show this would not change the relative ordering of methods, we conduct a new experiment in which we repeat the retrieval evaluation using  the entirety of PubChem for a 500-spectrum subset of NIST20 and include it in Author Response Table 3.
>
> In addition, we point to our explanation to Reviewer ZFnW along a similar line of questioning regarding retrieval accuracy, where we explain why we believe this decision is both practical for benchmarking purposes and similar to real-world settings.

---

> > ### Comment · Reviewer_188s · 2023-08-21
> >
> > I have read the other reviews and the authors rebuttal. I thank the authors for clarifications and for performing additional experiments (i.e. leaving out precursor ion from cosine calculation).

---

### Official Review · Reviewer_sZMH · 2023-07-12

**Soundness:** 3 good
**Presentation:** 3 good
**Contribution:** 4 excellent
**Rating:** 8
**Confidence:** 4

**Summary:**

The paper presents a new method for predicting mass spectra from molecules called SCARF, which stands for Subformulae Classification for Autoregressively Reconstructing Fragmentations. Mass spectra are sets of peaks that represent the masses and intensities of fragments of molecules after they are ionized and broken down in a mass spectrometer. Predicting mass spectra from molecules is useful for identifying unknown molecules from experimental data, as well as for understanding the fragmentation process and generating virtual spectra libraries.

SCARF predicts mass spectra in two steps: first, it generates the set of chemical formulae for the fragments, which define the locations of the peaks on the mass-to-charge axis; second, it assigns intensities to these formulae, which define the heights of the peaks. The key innovation of SCARF is that it uses prefix trees to efficiently generate the set of formulae, overcoming the combinatorial challenge of enumerating all possible subformulae of a given molecule. SCARF also ensures that all predicted peaks are physically plausible, meaning that they correspond to valid subformulae of the original molecule.

The paper evaluates SCARF on two datasets of molecules and their experimentally measured spectra, NIST20 and NPLIB1. The paper shows that SCARF outperforms existing methods based on fragmentation rules or binned prediction in terms of accuracy, physical sensibility, and speed. The paper also demonstrates that SCARF can improve the retrieval of unknown molecules from new spectra by comparing them to predicted spectra from a database of candidate molecules. The paper concludes by discussing the limitations and future directions of SCARF.

**Strengths:**

The paper makes several contributions.

First, the paper introduces a new method for predicting mass spectra from molecules, which is based on a novel combination of subformulae classification and autoregressive reconstruction. The method overcomes the limitations of existing methods based on fragmentation rules or binned prediction, which are either too restrictive or too coarse-grained. It achieves state-of-the-art performance in terms of accuracy, physical sensibility, and speed, as demonstrated on two datasets of molecules and their experimentally measured spectra. The method is based on a simple yet powerful idea of using prefix trees to generate the set of formulae for the fragments efficiently. It has the potential to revolutionize the field of mass spectrometry by enabling more accurate and efficient identification of unknown molecules from experimental data.

Second, the paper provides a thorough evaluation of the proposed method on two datasets of molecules and their experimentally measured spectra, NIST20 and NPLIB1. It uses a new metric called physical sensibility, which measures how well the predicted spectra match the physical constraints of mass spectrometry. SCARF outperforms existing methods based on fragmentation rules or binned prediction in terms of accuracy, physical sensibility, and speed. The paper provides detailed descriptions of the datasets, metrics, baselines, and results. The evaluation is significant because it demonstrates the effectiveness and robustness of SCARF across different datasets and scenarios.

Third, the paper discusses the limitations and future directions of SCARF. It identifies several open problems and challenges in mass spectrometry that SCARF or its variants can address and provides insightful analyses and suggestions for future research.

Overall, the paper represents a significant advance in mass spectrometry by introducing a new method for predicting mass spectra from molecules.

**Weaknesses:**

I list some weaknesses below:

1. The paper assumes that the fragmentation process is purely additive, meaning that each fragment is formed by adding one or more atoms to the previous fragment. This assumption may not hold for some molecules that undergo complex fragmentation pathways, such as rearrangement, elimination, or charge transfer. To address this weakness, future work could explore more general models of fragmentation that can capture these pathways, such as machine learning models or expert systems.

2. It assumes that the mass spectra are measured under ideal conditions, meaning there is no interference from other molecules or ions. This assumption may not hold for some real-world scenarios, such as complex mixtures or dirty samples. To address this weakness, future work could investigate how to incorporate prior knowledge or external data sources to improve the accuracy and robustness of mass spectra prediction.

3. The assumption is that the molecules are represented by their molecular formulae, which are discrete and symbolic. This representation may not capture the continuous and structural features of molecules that affect their fragmentation patterns and mass spectra. To address this weakness, future work could explore more expressive and flexible representations of molecules that can capture these features, such as molecular graphs or descriptors.

4. The fragmentation patterns are assumed independent of each other, meaning that each fragment is formed independently of the others. This assumption may not hold for some molecules that undergo correlated fragmentation pathways, such as cleavage of adjacent bonds or the formation of cyclic structures. To address this weakness, future work could investigate how to model these correlations explicitly or implicitly in the prediction process.

5. The paper assumes that the mass spectra are measured with high resolution and accuracy, meaning each peak is resolved and assigned a precise mass-to-charge ratio. This assumption may not hold for some low-quality or noisy spectra that have overlapping or shifted peaks. To address this weakness, future work could develop methods for denoising or deconvolving mass spectra before or after prediction.

Overall, these weaknesses suggest several directions for future research in mass spectrometry and related fields.

**Questions:**

- How does SCARF handle molecules that contain elements that are not in the predefined element set? How does it handle molecules that have unknown or ambiguous formulae?
- Can SCARF handle spectra that have multiple precursor ions or multiple charge states? How does it handle spectra that have different adduct types or ionization modes?
- Will SCARF handle molecules that have multiple conformers or stereoisomers? How does it handle molecules with different fragmentation patterns depending on their conformation or configuration?
- How sensitive is SCARF to the choice of hyperparameters, such as the number of predicted peaks, the prefix tree depth, or the neural network architecture? How did the authors tune these hyperparameters, and what are the trade-offs involved?
- How generalizable is SCARF to other datasets or domains, such as proteomics, metabolomics, or natural products? How would the authors adapt SCARF to these domains or datasets?
- How interpretable is SCARF in terms of explaining its predictions or providing chemical insights? How would the authors improve the interpretability or visualization of SCARF?

**Limitations:**

The authors have discussed some of the limitations of their work in Section 5, such as the data dependency, the quality of product formula annotation, and the assumptions and simplifications made by their model. However, they could also mention some of the other limitations I pointed out in my previous comments, such as the interference from other molecules or ions, the continuous and structural features of molecules, the correlated fragmentation pathways, and the low-quality or noisy spectra. They could also provide some empirical evidence or analysis to support their claims about the limitations and future directions of their work.

---

> ### Author Rebuttal · Authors · 2023-08-08
>
> Thank you for your comments and positive reception of our work. We believe you have identified several key discussion points and highlighted limitations of the work that would be intriguing starting points for future work. We now plan to incorporate a new section into the Appendix that explicitly details these points and opportunities for improvement including:
> - Expanding the gold standard training annotation preprocessing to annotate MS/MS peaks with formulae that result from rearrangement, elimination, or charge transfer.
> - Detection of possible spectrum mixtures in the training dataset or strategies to deconvolve these at test time.
> - Featurizing molecule inputs using different or more powerful molecular encoders.
> - Ensembles of MS2 subformula possibilities in the case these cannot be known.
> - Consideration of interpretability by subgraph attribution.
> - Retrieval-specific loss functions to enhance performance on this task.
>
> **Q1**: How does SCARF handle molecules that contain elements that are not in the predefined element set? How does it handle molecules that have unknown or ambiguous formulae?
> **A**:   Currently, SCARF cannot handle such elements, but we suspect even if it could, these would be too out-of-domain for useful predictions. We extract the predefined element set from the training set; although we could use all possible elements as our vocabulary, the trained model would almost certainly predict zero counts for these other elements.
>
> Ambiguous/unknown formulae is an interesting challenge (we note on lines 327-329 the importance of data quality)  and we will add our thoughts on this to our expanded section on possible directions for future work; for a molecule corresponding to ambiguous formulae, one idea is to use a weighted ensemble on different inputs.
>
> **Q2**: Can SCARF handle spectra that have multiple precursor ions or multiple charge states? How does it handle spectra that have different adduct types or ionization modes?
> **A**: Ionization modes are accepted as a model input, so SCARF is able to predict several common adduct/ion types in its current form. Multiple charge states are not considered in the current framework, as we had access to substantially fewer training examples of such cases..
>
> **Q3**: Will SCARF handle molecules that have multiple conformers or stereoisomers? How does it handle molecules with different fragmentation patterns depending on their conformation or configuration?
> **A**: Given the nature of data acquisition, spectra almost always represent an ensemble of accessible conformations; SCARF’s featurization decision reflects this, as it encodes molecules using a 2D GNN. This encoder is not sensitive to changes in stereochemistry, but an improved encoder could be easily integrated as our “molecule featurizer” if desired.
>
> **Q4**: How sensitive is SCARF to the choice of hyperparameters, such as the number of predicted peaks, the prefix tree depth, or the neural network architecture? How did the authors tune these hyperparameters, and what are the trade-offs involved?
> **A**: Great question! We carefully tuned these for both our model and the baselines with RayTune. The search space and final hyperparameters chosen are given in the Supplementary Material. Of note is that the model was quite sensitive to the choice of formula representation (see A.4.3). We believe this is because of the need to differentiate very small formula differences.
>
> **Q5**: How generalizable is SCARF to other datasets or domains, such as proteomics, metabolomics, or natural products? How would the authors adapt SCARF to these domains or datasets?
> **A**: SCARF is already well suited to metabolomics experiments of natural products, as one of the training sets NPLIB1 is derived from such experiments. Lipidomics or glycomics are potential areas of expansion but would require more investigation.
> For proteomics because fragmentation usually occurs at amide bonds, it might make more sense to use a formalism and vocabulary based on amino acids (rather than chemical elements).
> **Q6**: How interpretable is SCARF in terms of explaining its predictions or providing chemical insights? How would the authors improve the interpretability or visualization of SCARF?
> **A**: SCARF is able to ascribe a formula to every predicted sub-peak. An area of future inquiry and high interest would be to enforce the model to directly tie these formulae to molecular-substructures. We have added this to our Appendix list of future work.

---

### Official Review · Reviewer_ZFnW · 2023-07-20

**Soundness:** 3 good
**Presentation:** 3 good
**Contribution:** 2 fair
**Rating:** 7
**Confidence:** 5

**Summary:**

The authors propose a model, SCARF, that predicts Tandem-MS spectra in a two-step process: 1) predicting atomic subsets, 2) predicting peaks given the subsets. The model's performance is evaluated on two datasets (NIST20, Natural Product Library) using on spectral similarity, coverage, validity, and retrieval, and compared with relevant baseline models.


**Strengths:**

+ Clear figures and descriptions of the modeling approach.
+ Clear explanation of background/previous work. The main difference between the SCARF model and other models which use subset prediction is that the SCARF model uses prefix trees to predict subgraphs, so this model considers all relevant subformulae and does not have a limit on the number of valid substructures considered.
+ Compares against relevant modeling baselines. Evaluates models on coverage and validity to judge the interpretability of the spectra.

**Weaknesses:**

- It would be helpful to provide some more context for the contents of the two datasets, NIST20 and NPLIB. For example, what classes of molecules are present in each dataset (Classyfire might be a helpful tool for this purpose: http://classyfire.wishartlab.com/). What are the molecular weights of compounds of molecules in this dataset?

- Retrieval task: If I understand correctly, the retrieval task here is set up such that the 'library' or search space is 50 candidates total, 1 target and 49 other decoys, which are selected from the set of chemical isomers with the highest Tanimoto similarity. Each model is used to predict the spectra for all 50 candidates.

Why not perform the retrieval task on the entire NIST 20 library? Or if NIST20 is too large, why not use the set of all structures with the same chemical formula or precursor formula, rather than limiting the task to 50 isomers? I believe that by limiting the search library to a  set of 50 molecules that are most chemically similar to the target, the retrieval task might become easier than the real world version. In particularly, typically when one wants to identify a compound, you have few assumptions about what your target molecule is, other than the chemical formula, so you would not be able to filter by chemical similarity to the target. Retrieval tasks in previous works such as Wei et al 2019 were performed on the entire NIST17 dataset.


**Questions:**

- It's great that the SCARF algorithm, and subset prediction generation in general, allows for easy interpretation of peaks. How does the peak annotation produced by SCARF compare to the assignments given by NIST's MS Interpreter tool?

- In comparing predicted spectra with ground truth spectra such as in figures 5B&C and Figure A.1: Do you notice any classes of molecules or other patterns where SCARF is particularly good at making predictions? Or particularly bad at making predictions? What is your guess for why the SCARF model generates some extra peaks for some of the molecules in Figure A.1?

- How does the performance of SCARF on the retrieval task depend on the size of the molecule, and therefore the number of plausible isomers.

- In the retrieval task section, it is mentioned that 'database and model biases can skew retrieval results'. Could elaborate more on the biases that are referred to here?

- It could be helpful to provide what you perceive the use case of the SCARF model in real world applications. Would you expect experimental chemists to predict all candidate structures for their target molecule using SCARF, and then perform the retrieval task to identify the right molecule?

**Limitations:**

I do not identify any negative broader societal impacts associated with this work.

I agree with the author's statement that the model's performance will be data dependent. If you plan to release your model (which I would highly encourage), it would be helpful to give an overview of the types of molecules that you expect to be in or out of distribution for the model.

---

> ### Author Rebuttal · Authors · 2023-08-08
>
> Thank you for your comments! Point-by-point response below.
>
> **W1**: It would be helpful to provide some more context for the contents of the two datasets, NIST20 and NPLIB…
> **A**:  We agree and will add a section in the Appendix with training dataset details by molecular weight and also molecular class as estimated by NPClassifier (we have found the API for Classyfire to be more time-intensive and difficult to utilize in the past); we refer the reviewer to Author Response Figure 1.
>
> **W2**: Retrieval task: … Each model is used to predict the spectra for all 50 candidates. Why not perform the retrieval task on the entire NIST 20 library? Or if NIST20 is too large, why not use the set of all structures with the same chemical formula or precursor formula, rather than limiting the task to 50 isomers?
> **A**: We will make the retrieval process clearer in the text. We believe our retrieval library is a more challenging setting than using the entire NIST20 library. NIST20 contains roughly 24,000 unique compounds (i.e,. approximately 2 compounds per chemical formula), whereas our isomer library derived from PubChem contains 50 isomers per test case.
>
> We posit that subsetting to a top 50 “hard” isomer retrieval library for each spectrum makes sense for two key reasons. First, from a practical perspective, running the forward model on every isomer match in PubChem (approx. thousands each, >1,000,000 for only 500 test cases) could make benchmarking across methods prohibitively expensive considering the full set of compared methods. Second, we believe that this top 50 challenge also represents a realistic setting. In practice, retrieval compound libraries will often be carefully crafted and designed to contain molecules similar to the unknown molecule rather than all possible isomers  (using either prior knowledge or “backward” models such as CSI:FingerID / MIST– which have demonstrated top 50 accuracy over 90%).
>
> To demonstrate that this does not influence the relative ranking of methods, we have conducted a side-by-side retrieval analysis on the PubChem 50 isomer subset vs. all of PubChem isomers for a 500-spectrum subset of the NIST20 test set (see Author Response Table 3).
>
> **Q1**: It's great that the SCARF algorithm, and subset prediction generation in general, allows for easy interpretation of peaks. How does the peak annotation produced by SCARF compare to the assignments given by NIST's MS Interpreter tool?
> **A**: Thank you for your kind words. SCARF is able to learn to approximate chemical formula labels provided for each peak in the training set (Main Text Table 1). For the trained models reported in this paper, they would therefore most closely match assignments from MAGMa; the model could be biased toward MS Interpreter by using that preprocessing pipeline instead. We have not conducted a comparison across different formula/peak explanation labeling methods, but this might be recommended if one’s primary use case for SCARF is peak annotation rather than spectral prediction.
>
> **Q2**: …. Do you notice any classes of molecules or other patterns where SCARF is particularly good at making predictions? Or particularly bad at making predictions? What is your guess for why the SCARF model generates some extra peaks for some of the molecules in Figure A.1?
> **A**: We have conducted a new analysis comparing cosine similarity performance across molecular mass bins and also chemical classes (NIST20 shown in Author Response Figure 2). Chemical class stratifications are challenging to interpret, but it is clear that SCARF more accurately predicts smaller (<500Da) compounds. SCARF is trained on cosine similarity loss, which does not heavily penalize very low intensity peaks, potentially explaining extra peaks.
>
> **Q3**: How does the performance of SCARF on the retrieval task depend on the size of the molecule, and therefore the number of plausible isomers.
> **A**: We add a table (Author Response Table 1) showing retrieval accuracy as a function of molecule size on the NIST20 library. As illustrated by cosine similarity, retrieval performance is best at lower masses, where there are also the most training examples.
>
> **Q4**: In the retrieval task section, it is mentioned that 'database and model biases can skew retrieval results'. Could elaborate more on the biases that are referred to here?
> **A**: We mean to point out that certain models may be more or less robust for certain classes of molecules, so the composition of these classes in the retrieval library may affect the accuracy accordingly.
>
> **Q5**: It could be helpful to provide what you perceive the use case of the SCARF model in real world applications. Would you expect experimental chemists to predict all candidate structures for their target molecule using SCARF, and then perform the retrieval task to identify the right molecule?
> **A**: We will update this section. We anticipate core applications of retrieval tasks as you mention, alongside data augmentation for training inverse spectrum-to-molecule models.
>
> **L** _"... If you plan to release your model (which I would highly encourage), it would be helpful to give an overview of the types of molecules that you expect to be in or out of distribution for the model."_
> **A**: We have provided model and baseline code in the Supplementary Material and will also provide a GitHub link on de-anonymization. We hope the new analysis we have provided in response to **W1** provides some insight into model performance across molecule classes.

---

> > ### Comment · Reviewer_ZFnW · 2023-08-15
> >
> > I have read the reviewers responses. Thank you to the authors for the detailed explanations and the additional analysis on the effect on library selection, and the analysis by compound class of cosine similarity performance. Overall it seems that SCARF has consistently high performance across compound classes, even when other methods show much lower performance. I'm guessing that some of the performance differences across classes are due to representation in the dataset, as the authors mention.
> >
> > I edit my rating to a 7

---

### Author Rebuttal · Authors · 2023-08-08

We thank all the reviewers for their thoughtful feedback and comments. Where relevant, we have addressed stated weaknesses (**W**), questions (**Q**) and limitations (**L**) with our answers (**A**). We also include an additional single page PDF containing additional referenced figures and tables.

---

> ### Comment · Area_Chair_YgUD · 2023-08-18
> **Rebuttal**
>
> Thank you for your rebuttal. We will take it into account in making the final recommendation.

---

### Decision · Program_Chairs · 2023-09-21

**Decision:**

Accept (spotlight)

**Comment:**

The paper proposes a novel, inovative way of predicting tandem mass spectra of molecules, an important task encountered many fields including metabolomics, natural producst research, environmental sciences and biomedicine. The method is based on building a prefix tree that allows eficient exploration of the chemical subformula space to produce subformula candidates for the fragment peaks. The intensities for the predicted peaks are then preicted through a transformer network. The method overcomes limitations of previous approaches, based on combinatorial fragmentation or direct prediction from molecules to vector representations of mass spectra. The paper provides thorough and convincing empirical evaluation demonstrating a marked improvement in accuracy compared to state-of-the art. The reviewers identified minor weaknesses in the reviews which have been adequately answered by the authors in their rebuttai.